# Ca²⁺ sensor synaptotagmin-1 mediates exocytosis in mammalian photoreceptors

Justin J Grassmeyer[1,2], Asia L Cahill[1], Cassandra L Hays[1,3], Cody Barta[1], Rolen M Quadros[4], Channabasavaiah B Gurumurthy[4,5], Wallace B Thoreson[1,2]*

[1]Truhlsen Eye Institute, Department of Ophthalmology and Visual Sciences, University of Nebraska Medical Center, Omaha, United States; [2]Department of Pharmacology and Experimental Neuroscience, University of Nebraska Medical Center, Omaha, United States; [3]Department of Cellular and Integrative Physiology, University of Nebraska Medical Center, Omaha, United States; [4]Mouse Genome Engineering Core Facility, Vice Chancellor for Research Office, University of Nebraska Medical Center, Omaha, United States; [5]Developmental Neuroscience, Munroe Meyer Institute for Genetics and Rehabilitation, University of Nebraska Medical Center, Omaha, United States

**Abstract** To encode light-dependent changes in membrane potential, rod and cone photoreceptors utilize synaptic ribbons to sustain continuous exocytosis while making rapid, fine adjustments to release rate. Release kinetics are shaped by vesicle delivery down ribbons and by properties of exocytotic Ca²⁺ sensors. We tested the role for synaptotagmin-1 (Syt1) in photoreceptor exocytosis by using novel mouse lines in which Syt1 was conditionally removed from rods or cones. Photoreceptors lacking Syt1 exhibited marked reductions in exocytosis as measured by electroretinography and single-cell recordings. Syt1 mediated all evoked release in cones, whereas rods appeared capable of some slow Syt1-independent release. Spontaneous release frequency was unchanged in cones but increased in rods lacking Syt1. Loss of Syt1 did not alter synaptic anatomy or reduce Ca²⁺ currents. These results suggest that Syt1 mediates both phasic and tonic release at photoreceptor synapses, revealing unexpected flexibility in the ability of Syt1 to regulate Ca²⁺-dependent synaptic transmission.
DOI: https://doi.org/10.7554/eLife.45946.001

*For correspondence: wbthores@unmc.edu

**Competing interests:** The authors declare that no competing interests exist.

## Introduction

Retinal rod and cone photoreceptors transform changes in light intensity into graded, non-spiking changes in membrane voltage ($V_m$). The membrane potential of photoreceptors in darkness is relatively depolarized, allowing Ca²⁺ to enter the cell continuously through voltage-gated Ca$_V$1.4 Ca²⁺ channels clustered beneath ribbon-style active zones. This in turn stimulates continuous Ca²⁺-dependent exocytosis of glutamate-filled vesicles (*LoGiudice and Matthews, 2009*; *Mercer and Thoreson, 2011*; *Pangrsic et al., 2018*). Absorption of photons in the photoreceptor outer segment causes photoreceptors to hyperpolarize, thereby reducing Ca²⁺ influx and the rate of exocytosis. Local Ca²⁺ changes at the base of ribbons are sensed by Ca²⁺-binding proteins embedded in the vesicular membrane that promote vesicle-cell membrane fusion. Various vesicular Ca²⁺-sensing proteins including synaptotagmin (Syt), otoferlin, and Doc2 are capable of mediating exocytosis in different neurons (*Kaeser and Regehr, 2014*; *Pang and Südhof, 2010*; *Pangršič et al., 2012*).

Different kinetic phases of exocytosis can be shaped by properties of the Ca²⁺ sensor mediating release. At many synapses, fast synchronous, slow asynchronous, and spontaneous modes of release utilize distinct Ca²⁺ sensors with differing Ca²⁺ binding properties (*Kaeser and Regehr, 2014*). Fast phasic release evoked by brief depolarization is typically mediated by Syt1 or the closely-related

Syt2 (*Augustine and Charlton, 1986*; *Bollmann et al., 2000*; *Dodge and Rahamimoff, 1967*; *Jarsky et al., 2010*; *Landò and Zucker, 1994*; *Schneggenburger and Neher, 2000*). Syt7 has been suggested to mediate a slow, asynchronous phase of release that follows the initial fast burst of release (*Bacaj et al., 2013*; *Luo et al., 2015*; *Turecek and Regehr, 2018*; *Wen et al., 2010*). Doc2 has been proposed as a sensor for both asynchronous and spontaneous release (*Groffen et al., 2010*; *Pang et al., 2011*; *Yao et al., 2011*).

Rod and cone photoreceptors are capable of fast and slow phases of depolarization-evoked release as well as spontaneous release (*Cadetti et al., 2005*; *DeVries and Schwartz, 1999*; *Rabl et al., 2005*). At ribbon synapses, the kinetics of release can be shaped not only by properties of the sensor molecules but also by the rate at which vesicles are delivered to release sites at the base of the ribbon (*Jackman et al., 2009*; *Matthews and Fuchs, 2010*). In cone photoreceptors, the initial fast phase of release matches the number of vesicles in the readily releasable pool that are tethered at the base of the ribbon in contact with the adjacent plasma membrane (*Bartoletti et al., 2010*). Sustained depolarization stimulates a second phase of release that matches the number of vesicles attached to the remainder of the ribbon, suggesting that this slower phase reflects release of vesicles that have descended to re-occupy vacant release sites at the ribbon base. After depleting these two pools of ribbon-attached vesicles, release from cones can be maintained indefinitely at a linear rate, presumably reflecting replenishment of the ribbon by cytoplasmic vesicles (*Bartoletti et al., 2010*). Ribbon-mediated release from rods appears to utilize similar mechanisms (*Li et al., 2010*; *Van Hook and Thoreson, 2015*), but rods are also capable of slow release at non-ribbon release sites that may be analogous to asynchronous release at conventional synapses (*Babai et al., 2010*; *Cadetti et al., 2006*; *Suryanarayanan and Slaughter, 2006*). Do the different kinetic phases of release in photoreceptors arise entirely from structural factors that shape the kinetics of vesicle delivery to the membrane or do they involve the use of distinct $Ca^{2+}$ sensors?

The identity of the molecular $Ca^{2+}$ sensors that mediate different forms of release in photoreceptors is unknown. Unlike release from bipolar cells and most other neurons, exocytosis from amphibian photoreceptor synapses shows an unusually high $Ca^{2+}$ affinity and shallow $Ca^{2+}$ cooperativity, suggesting photoreceptors may employ an unusual $Ca^{2+}$ sensor (*Duncan et al., 2010*; *Rabl et al., 2005*; *Rieke and Schwartz, 1996*; *Thoreson et al., 2004*). Many properties of release are similar in mammals and amphibians, but immunohistochemical studies suggest that mammalian photoreceptors express Syt1, the $Ca^{2+}$ sensor used at many conventional synapses, but amphibian photoreceptors do not (*Berntson and Morgans, 2003*; *Fox and Sanes, 2007*; *Heidelberger et al., 2003*; *Ullrich et al., 1994*). Syt2 is absent from both amphibian and mammalian photoreceptors. We tested a role for Syt1 in synaptic release from rods and cones by developing a conditional Syt1 knockout mouse line to remove Syt1 specifically from rods or cones. Electroretinography (ERG) experiments and single-cell recordings from rods and cones indicate that a single sensor, Syt1, mediates sustained and phasic release from rods and cones. Other $Ca^{2+}$ sensors may be involved in mediating some slow release from rods and spontaneous release from both rods and cones.

## Results

### Syt1 expression was specifically abolished in either rods or cones using conditional knockout mice

Mice lacking *Syt1* die within 48 hr of birth (*Geppert et al., 1994*) and the retina is not fully developed until P14. To probe the potential function of Syt1 in mouse photoreceptors, we therefore generated a conditional, *Cre*-dependent Syt1 knockout mouse line with *LoxP* sites flanking exon 6 of *Syt1* (*Quadros et al., 2017*) (*Figure 1A–B*). PCR experiments showed proper insertion of both *LoxP* sites (*Quadros et al., 2017*). 5' *LoxP* PCR results are illustrated in *Figure 1B*. This line was crossed with mice that express Cre recombinase specifically in rods under control of the rhodopsin promoter (*Rho-iCre*) (*Li et al., 2005*) or in cones under control of the human red-green pigment promoter (HRGP-*Cre*) (*Le et al., 2004*). To facilitate targeting of cones for single-cell recordings (described below), we also crossed HRGP-*Cre; Syt1* mice with a Cre-dependent tdTomato-expressing reporter line (Ai14).

To test for Syt1 protein in rods and cones, we examined immunohistochemical labeling for Syt1 using an antibody that labels the C2A domain. In control retinas, Syt1 immunoreactivity was evident

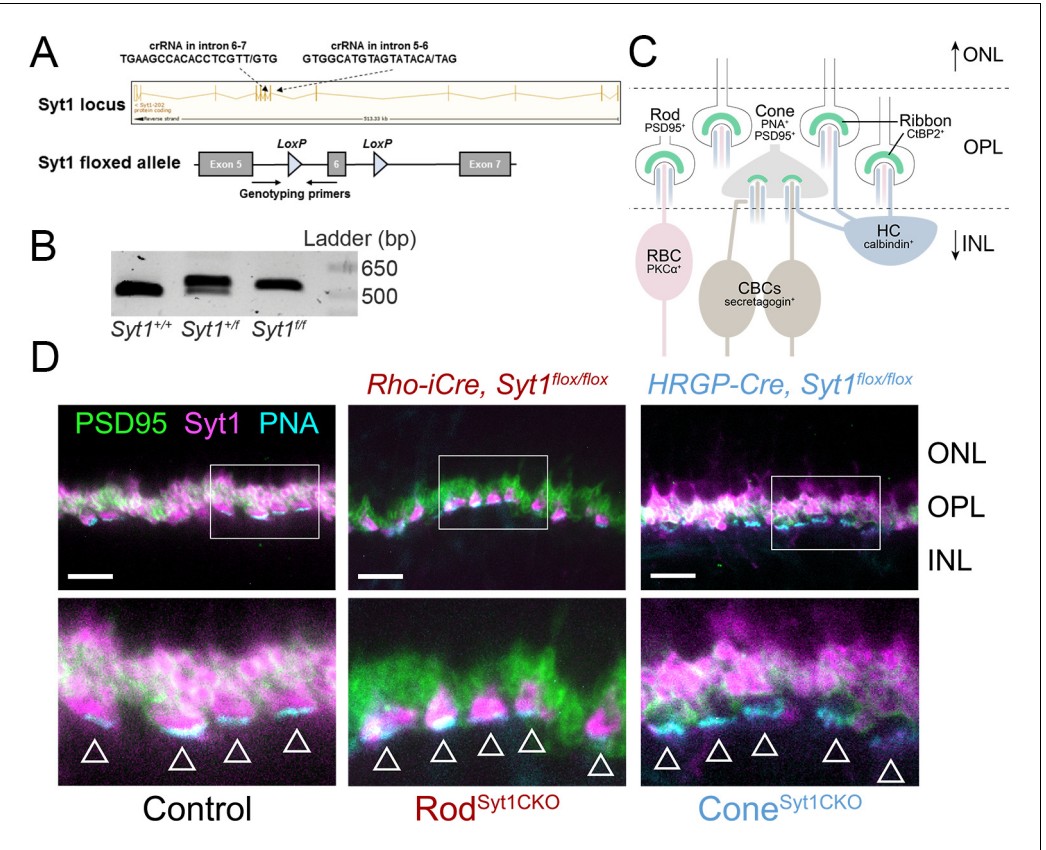

**Figure 1.** Syt1 was conditionally deleted from rods and cones in Rod[Syt1CKO] and Cone[Syt1CKO] retinas, respectively. (A) Top: Syt1 locus showing crRNA sequences used for inserting *LoxP* sites flanking exon 6; '/" in the amino acid sequence indicates the nucleotide positions where *LoxP* sites were inserted. Bottom: schematic of the *Syt1[flox]* allele showing location of genotyping primers and *LoxP* sites. (B) 5′ LoxP PCR of the *Syt1* allele from WT (*Syt1[+/+]*) and *Syt1* floxed mice (*Syt1[+/f]*: heterozygous, *Syt1[f/f]*: homozygous). Expected band sizes are 484 bp for the WT allele and 524 bp for the floxed allele. (C) Diagram illustrating fluorescent labels used for different cell types. Rod and cone terminals can be labeled with antibodies to PSD95. The base of cone terminals can be labeled with fluorescently-conjugated peanut agglutinin (PNA). Rod and cone ribbons were labeled with antibodies to CtBP2. Horizontal cells (HCs), rod bipolar cells (RBCs), and cone bipolar cells (CBCs) were labeled with antibodies to calbindin, PKCα, and secretagogin, respectively. (D) Images of control, Rod[Syt1CKO], and Cone[Syt1CKO] retinas labeled with PNA (cyan) to mark cone terminals as well as antibodies to PSD95 (green) and Syt1 (magenta). Bottom images show magnified regions outlined in the top images. Arrowheads indicate cone terminals. Scale bars = 10 μm. ONL: outer nuclear layer, INL: inner nuclear layer.
DOI: https://doi.org/10.7554/eLife.45946.002

in both rod and cone terminals in the outer plexiform layer (OPL) (*Figure 1D*), consistent with previous studies (*Berntson and Morgans, 2003*; *Fox and Sanes, 2007*; *von Kriegstein and Schmitz, 2003*). Syt1 immunoreactivity in *Syt1* heterozygote retinas (*Rho-iCre; Syt1[+/flox]* and HRGP-*Cre; Syt1[+/flox]*) was indistinguishable from controls (*Syt1[flox/flox]* or *Cre[+], Syt1[+/+]*; data not shown). As illustrated by the diagram in *Figure 1C*, PSD95 was used to label presynaptic rod and cone terminals in the OPL (*Koulen et al., 1998*) while fluorescently-conjugated peanut agglutinin (PNA) was used to label the base of cone terminals (*Blanks and Johnson, 1984*). Cone terminals are indicated by arrowheads in *Figure 1D*. In contrast to control retinas, *Rho-iCre; Syt1[flox/flox]* homozygous mutant (hereafter called Rod[Syt1CKO]) retinas exhibited robust Syt1 expression in cone terminals but Syt1 was completely absent from rod spherules (*Figure 1D*, middle). Conversely, in HRGP-*Cre; Syt1[flox/flox]* homozygous mutant (hereafter called Cone[Syt1CKO]) retinas, Syt1 was absent from cone terminals but strongly expressed in rod spherules (*Figure 1D*, right). These results confirm that Syt1 is expressed

robustly in mouse photoreceptors and show that Syt1 expression was abolished specifically from rods and cones in Rod[Syt1CKO] and Cone[Syt1CKO] retinas, respectively.

## Removal of Syt1 from photoreceptors diminishes ERG b-waves

We first examined the functional impact of the absence of Syt1 from rods and cones by evaluating light-evoked ERG responses using an ex vivo eyecup preparation (*Newman and Bartosch, 1999*). We focused on the ERG a-wave, a negatively-polarized wave that reflects the hyperpolarizing light responses of rod and cone photoreceptors, and the b-wave, a positively-polarized wave that reflects the depolarization of ON bipolar cells arising from the light-evoked cessation of tonic glutamate release from photoreceptors. The bath solution was supplemented with 100 µM $BaCl_2$ to block the Müller cell-mediated slow PIII component of the ERG (*Bolnick et al., 1979*). A-waves were measured from baseline to the negative going inflection. B-waves were measured from the trough of the a-wave to the peak of the positive-going b-wave (see arrows in *Figure 2A–B*). Using brief (20 ms) flashes without background illumination, control retinas exhibited a b-wave intensity-response function composed of rod-driven responses at low intensities and a mixture of rod- and cone-driven activity at higher intensities (*Figure 2A–B*). *Figure 2A* shows example responses evoked by a low intensity light flash ($10^{-4}$ of maximum) in control, Rod[Syt1CKO], and Cone[Syt1CKO] retinas. The small a-wave evoked at this intensity is obscured by the b-wave so only the b-wave is evident. *Figure 2B* shows responses to a bright flash ($10^{-4}$ of maximum) that evokes large a- and b-waves in control mice. In control retinas, the b-wave became significantly non-zero at a flash intensity $10^{-5}$ of maximum (p=0.003, one-sample t-test). Rod[Syt1CKO] retinas, in which rods lack Syt1, exhibited markedly diminished or no b-wave activity at the dimmest flash intensities; a much higher flash intensity was required for Rod[Syt1CKO] b-waves to become significantly non-zero (p>0.05 at intensities below $10^{-3}$ of maximum; *Figure 2B*, maroon data). Conversely, Cone[Syt1CKO] retinas exhibited rod-mediated b-waves at low intensities that were as sensitive as controls; the threshold for b-wave detection was the same as controls ($10^{-5}$ of maximum intensity; p=0.004; *Figure 2B*, blue data). However, b-wave amplitudes from Cone[Syt1CKO] retinas did not increase to the amplitude of control b-waves when evoked by higher intensity flashes, suggesting a diminished cone contribution to b-waves in Cone[Syt1CKO] retinas. The effect of Syt1 deletion from cones was partially masked by increasing rod-driven responses at bright flash intensities (*Ronning et al., 2018*).

While the b-waves were reduced, there was no reduction in a-waves of Rod[Syt1CKO] or Cone[Syt1CKO] retinas throughout the response range (*Figure 2C*). The threshold at which a-waves became significantly non-zero was at an intensity $10^{-3}$ of maximum for both control and Rod[Syt1CKO] retinas while Cone[Syt1CKO] retinas became significantly non-zero at an intensity $10^{-3.5}$ of maximum. A-waves are only evident with relatively bright flashes because they are masked at dimmer intensities by the b-wave. To evaluate the impact of Syt1 deletion on rod photoresponses at lower intensities, photoreceptor responses were isolated from other components of the ERG by using 20 µM L-AP4 (along with 100 µM $BaCl_2$) to eliminate the b-wave (*Stockton and Slaughter, 1989*). We examined isolated photoreceptor responses in control and Rod[Syt1CKO] retinas (*Figure 2D*). At the lowest intensities ($\leq 10^{-4}$ attenuation), we used a light flash of 500 ms rather than 20 ms to make it easier to measure rod responses. There was a significant variation of responses by light intensity as expected (2-way ANOVA, p<0.0001) but not by genotype (p=0.095). There was also no significant interaction between intensity and genotype (i.e., light did not affect the genotypes differently, p=0.25). These results indicate that diminished photoresponses were not responsible for the marked b-wave reductions in Rod[Syt1CKO] and Cone[Syt1CKO] retinas.

Because the b-wave reflects bipolar cell responses and the a-wave reflects photoreceptor responses, the ratio of the b-wave to a-wave provides a measure of the efficiency of translating light-evoked $V_m$ changes in photoreceptors into a change in neurotransmission, with higher ratios indicating a more efficient transformation. We hypothesized that if control photoreceptors exhibited a greater rate of continuous exocytosis, they would exhibit larger b-wave/a-wave ratios. At the brightest three intensities where a- and b-waves were reliably attained, Rod[Syt1CKO] and Cone[Syt1CKO] retinas were both significantly less efficient than controls at two of the three intensities, suggesting that Rod[Syt1CKO] and Cone[Syt1CKO] retinas both exhibited less tonic glutamate release in darkness (*Figure 2E*).

Two alternate approaches were used to further test the role of Syt1 in mediating ERG responses. First, we delivered two flashes in rapid succession without background illumination (*Figure 3A–B*).

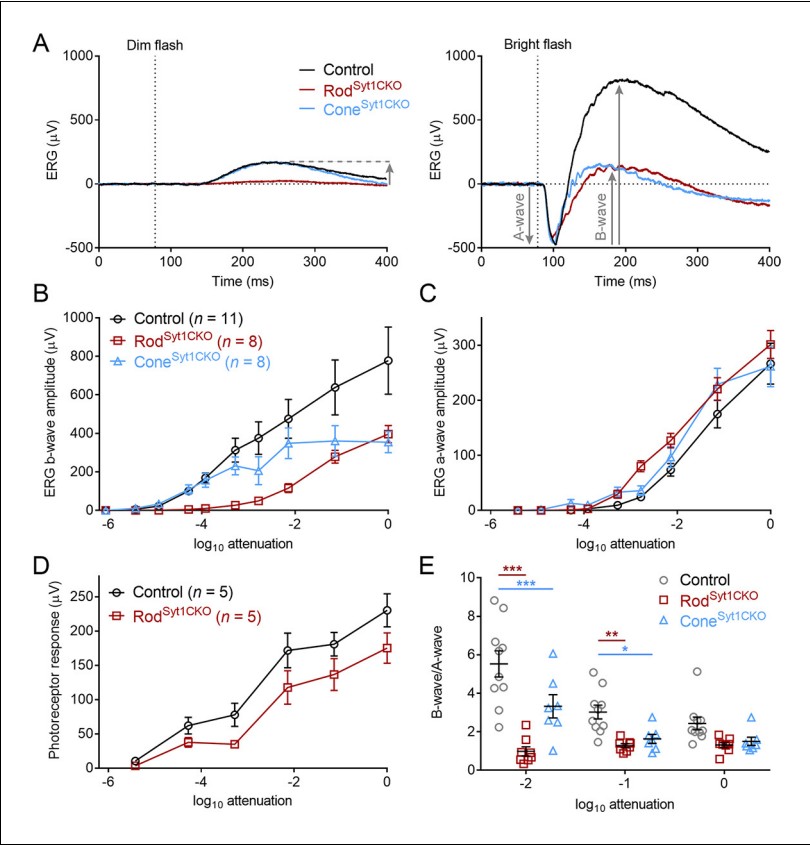

**Figure 2.** Flash ERG responses suggest a significant role for Syt1 in photoreceptor neurotransmission. (**A**) Example ERG recordings from control, Rod$^{Syt1CKO}$, and Cone$^{Syt1CKO}$ retinas in response to 20 ms dim (left, $10^{-4}$ of maximum) and bright (right, $10^{-1}$ of maximum) flashes. Downward gray arrows indicate a-waves and upward arrows indicate b-waves (a-waves are obscured by b-waves at low intensities). (**B**) Average b-wave flash intensity-response functions for the three genotypes. Maximum intensity b-waves: control: 777.7 ± 174.4 µV, Rod$^{Syt1CKO}$: 396.1 ± 44.3 µV, Cone$^{Syt1CKO}$: 354.0 ± 55.5 µV; Rod$^{Syt1CKO}$ vs. control: p=0.10, Cone$^{Syt1CKO}$ vs. control: p=0.06 (one-way ANOVA). (**C**) A-wave flash intensity-response curves for the three genotypes. Same legend as B). Maximum intensity a-waves: control: 266.7 ± 37.1 µV, Rod$^{Syt1CKO}$: 301.4 ± 25.4 µV, Cone$^{Syt1CKO}$: 261.6 ± 37.0 µV; Rod$^{Syt1CKO}$ vs. control: p=0.72, Cone$^{Syt1CKO}$ vs. control: p=0.99 (one-way ANOVA). (**D**) Isolated photoreceptor responses in control and Rod$^{Syt1CKO}$ retinas. No means were significantly different (t-tests corrected for multiple comparisons). (**E**) B-wave/a-wave ratios for control (n = 10), Rod$^{Syt1CKO}$ (n = 8), and Cone$^{Syt1CKO}$ (n = 7) retinas. $10^{-2}$ attenuation: control: 5.5 ± 0.68, Rod$^{Syt1CKO}$: 0.99 ± 0.24, Cone$^{Syt1CKO}$: 3.3 ± 0.61; Rod$^{Syt1CKO}$ vs. control: p=0.0001, Cone$^{Syt1CKO}$ vs. control: p=0.0005. $10^{-1}$ attenuation: control: 3.0 ± 0.36, Rod$^{Syt1CKO}$: 1.3 ± 0.11, Cone$^{Syt1CKO}$: 1.6 ± 0.23; Rod$^{Syt1CKO}$ vs. control: p=0.004, Cone$^{Syt1CKO}$ vs. control: p=0.03. Maximum intensity: control: 2.4 ± 0.33, Rod$^{Syt1CKO}$: 1.3 ± 0.13, Cone$^{Syt1CKO}$: 1.5 ± 0.22; Rod$^{Syt1CKO}$ vs. control: p=0.08, Cone$^{Syt1CKO}$ vs. control: p=0.19 (repeated measures 2-way ANOVA). ***p≤0.0005, **p=0.004, *p=0.03.

DOI: https://doi.org/10.7554/eLife.45946.003

Responses evoked by the first flash reflected the summed activity of both rods and cones, but cones dominated responses to the second flash because the initial flash desensitizes rods. The ratio of the two b-waves (b-wave$_2$/b-wave$_1$) reflects the amount of desensitization and was used to estimate cone contributions to the b-wave. As expected, desensitization increased (i.e., b-wave$_2$/b-wave$_1$ ratio decreased) as flash intensity increased, regardless of genotype (*Figure 3C*). We predicted that if rod neurotransmission was reduced in Rod$^{Syt1CKO}$ retinas, responses would be more cone-dominated and therefore exhibit less desensitization than controls. Conversely, we predicted that responses of Cone$^{Syt1CKO}$ retinas would be more rod-dominated and exhibit increased desensitization. Consistent with these predictions, Rod$^{Syt1CKO}$ b-wave$_2$/b-wave$_1$ ratios were increased significantly and Cone$^{Syt1CKO}$ b-wave$_2$/b-wave$_1$ ratios were decreased significantly relative to control retinas at all intensities (*Figure 3C*). The degree of a-wave desensitization did not differ among the

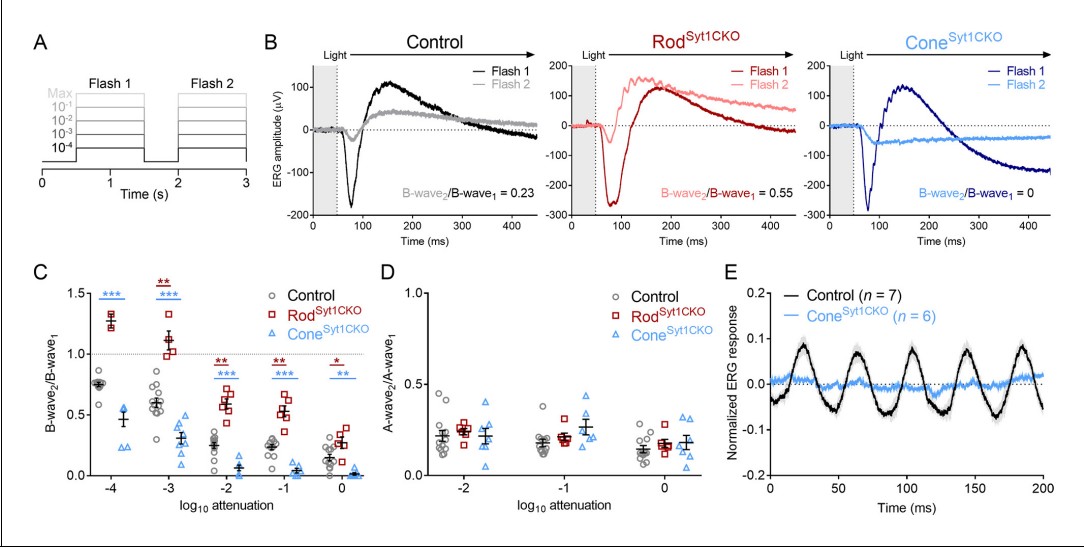

**Figure 3.** Double-flash and flicker ERG responses indicate severely impaired neurotransmission in photoreceptors that lack Syt1. (**A**) Schematic of the double-flash ERG protocol used for data in B-D. (**B**) Examples of superimposed ERG responses to the first and second flashes elicited by the protocol in A ($10^{-1}$ of maximum intensity) from control, Rod$^{Syt1CKO}$, and Cone$^{Syt1CKO}$ retinas. (**C**) B-wave$_2$/B-wave$_1$ ratios for the three genotypes across flash intensities. $10^{-4}$ attenuation: control: $0.75 \pm 0.02$, $n = 12$; Rod$^{Syt1CKO}$: $1.27 \pm 0.06$, $n = 2$; Cone$^{Syt1CKO}$: $0.46 \pm 0.06$, $n = 7$; Rod$^{Syt1CKO}$ vs. control not analyzed because $n = 2$, Cone$^{Syt1CKO}$ vs. control: p=0.0005. $10^{-3}$ attenuation: control: $0.60 \pm 0.04$, $n = 13$; Rod$^{Syt1CKO}$: $1.11 \pm 0.08$, $n = 4$; Cone$^{Syt1CKO}$: $0.31 \pm 0.05$, $n = 8$; Rod$^{Syt1CKO}$ vs. control: p=0.0005, Cone$^{Syt1CKO}$ vs. control: p=0.0008. $10^{-2}$ attenuation: control: $0.25 \pm 0.02$, $n = 13$; Rod$^{Syt1CKO}$: $0.59 \pm 0.04$, $n = 6$; Cone$^{Syt1CKO}$: $0.064 \pm 0.02$, $n = 7$; Rod$^{Syt1CKO}$ vs. control: p=0.0005, Cone$^{Syt1CKO}$ vs. control: p=0.0008. $10^{-1}$ attenuation: control: $0.24 \pm 0.02$, $n = 12$; Rod$^{Syt1CKO}$: $0.53 \pm 0.04$, $n = 6$; Cone$^{Syt1CKO}$: $0.042 \pm 0.02$, $n = 6$; Rod$^{Syt1CKO}$ vs. control: p=0.0005, Cone$^{Syt1CKO}$ vs. control: p=0.0008. Maximum intensity: control: $0.15 \pm 0.03$, $n = 12$; Rod$^{Syt1CKO}$: $0.27 \pm 0.05$, $n = 5$; Cone$^{Syt1CKO}$: $0.014 \pm 0.01$, $n = 7$; Rod$^{Syt1CKO}$ vs. control: p=0.02, Cone$^{Syt1CKO}$ vs. control: p=0.005 (one-way ANOVAs corrected for multiple comparisons). (**D**) A-wave$_2$/A-wave$_1$ ratios for the three genotypes. No comparisons with control were significant at any intensity. $10^{-2}$ attenuation: control: $0.22 \pm 0.03$, $n = 13$; Rod$^{Syt1CKO}$: $0.24 \pm 0.02$, $n = 6$; Cone$^{Syt1CKO}$: $0.22 \pm 0.04$, $n = 7$; Rod$^{Syt1CKO}$ vs. control: p=0.93, Cone$^{Syt1CKO}$ vs. control: p=0.99. $10^{-1}$ attenuation: control: $0.18 \pm 0.02$, $n = 12$; Rod$^{Syt1CKO}$: $0.21 \pm 0.02$, $n = 6$; Cone$^{Syt1CKO}$: $0.27 \pm 0.04$, $n = 6$; Rod$^{Syt1CKO}$ vs. control: p=0.93, Cone$^{Syt1CKO}$ vs. control: p=0.17. Maximum intensity: control: $0.14 \pm 0.02$, $n = 12$; Rod$^{Syt1CKO}$: $0.18 \pm 0.02$, $n = 6$; Cone$^{Syt1CKO}$: $0.18 \pm 0.04$, $n = 7$; Rod$^{Syt1CKO}$ vs. control: p=0.93, Cone$^{Syt1CKO}$ vs. control: p=0.76 (one-way ANOVAs corrected for multiple comparisons). (**E**) Average ($\pm$ SEM) normalized ERG responses of control and Cone$^{Syt1CKO}$ retinas to a bright 25 Hz flicker stimulus. ***p<0.001, **p=0.005, *p=0.02.

DOI: https://doi.org/10.7554/eLife.45946.004

three groups, supporting the finding that phototransduction is normal in photoreceptors that lack Syt1 (*Figure 3D*). These results provide further evidence of significant neurotransmission deficits in rods and cones lacking Syt1.

B-waves evoked by the second bright flash of the double flash protocol in Cone$^{Syt1CKO}$ retinas were nearly undetectable (*Figure 3C*), suggesting that synaptic output from cones lacking Syt1 was abolished. As a further test of cone function, we recorded the ERG evoked by a 25 Hz on/off square-wave stimulus at a bright intensity. Unlike cones, rods are incapable of following rapidly flickering changes at this high frequency (*Tanimoto, 2009*) and quickly adapt to the mean luminance of the bright stimulus. ERGs from control mice were able to follow 25 Hz flickering stimuli, but ERGs from Cone$^{Syt1CKO}$ retinas were not (*Figure 3E*). The absence of any detectable flicker response provides further evidence that release from cones was eliminated by the loss of Syt1 in Cone$^{Syt1CKO}$ retinas. Together, the results in *Figure 2* and *Figure 3* demonstrate that Syt1 is required for continuous exocytosis from photoreceptors in darkness.

## Exocytosis from cones is markedly reduced in the absence of Syt1

The ERG results suggested a significant role for Syt1 in rod and cone neurotransmission, but ERG recordings reflect the simultaneous activity of entire neuronal populations. To probe Syt1 function on a cellular level, we measured synaptic release from individual rods and cones with whole-cell recordings. Exocytosis was assessed in two ways: 1) measuring the inhibition of $I_{Ca}$ produced by the

release of protons upon synaptic vesicle fusion, and 2) recording anion currents generated by the activity of presynaptic glutamate transporters.

We examined the transient inhibition of $I_{Ca}$ caused by the release of vesicular protons (*DeVries, 2001*; *Palmer et al., 2003*; *Vincent et al., 2018*) during exocytosis in cones. As described further below, $I_{Ca}$ is much smaller in rods than cones so $I_{Ca}$ inhibition was not apparent in rods. For these experiments, tdTomato-positive cones in retinal slices from littermate HRGP-$Cre^{tdT}$; $Syt1^{+/flox}$ (control) and HRGP-$Cre^{tdT}$; $Syt1^{flox/flox}$ (Cone$^{Syt1CKO}$) retinas were targeted for recording (*Figure 4A*). Depolarizing steps produced a robust inward $I_{Ca}$ in both control and Syt1-deficient cones, and immediately following $I_{Ca}$ activation, $I_{Ca}$ in control cones—but not cones lacking Syt1— exhibited a transient outward deflection similar to the vesicular proton-mediated inhibition of $I_{Ca}$ described previously in cones and bipolar cells (*DeVries, 2001*; *Palmer et al., 2003*) (*Figure 4B*). To confirm that the transient inhibition of $I_{Ca}$ in control cones was mediated by $H^+$, buffering of the extracellular solution was strongly increased with 20 mM HEPES, which eliminated $I_{Ca}$ inhibition (*Figure 4C*). To test whether the source of $H^+$ was exocytosis of low-pH vesicular contents, cones were depolarized with two test pulses separated by a brief interval (25 ms steps with a 50 ms inter-pulse interval). Repetitive stimulation in photoreceptors produces marked presynaptic depression due to the depletion of releasable vesicles (*Grabner et al., 2016*; *Innocenti and Heidelberger, 2008*; *Rabl et al., 2006*). $I_{Ca}$ inhibition was evident in control cones during the first step but not during the second step when the pool of releasable vesicles had been depleted (*Figure 4D*). Together, these results confirmed that the transient inhibition of $I_{Ca}$ during depolarizing steps in control cones was due to acidification of the cleft upon the release of presynaptic vesicles. By subtracting $I_{Ca}$

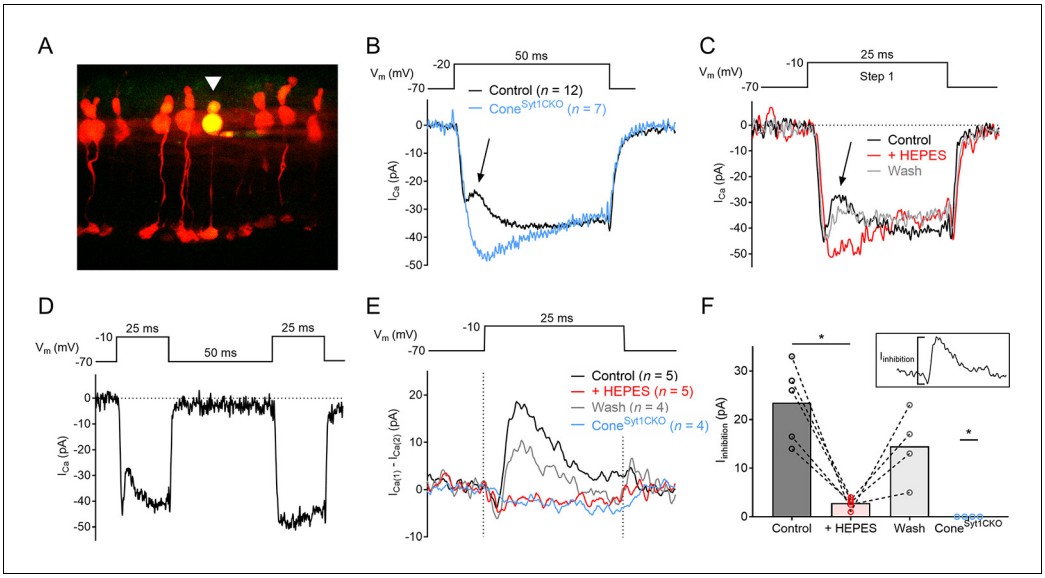

**Figure 4.** Cones lacking Syt1 do not experience inhibition of $I_{Ca}$ by vesicular $H^+$. (**A**) Image of tdTomato$^+$ cones in an ex vivo retinal slice. One cone was filled with a patch pipette solution supplemented with Lucifer yellow (arrowhead). (**B**) Average $I_{Ca}$ recorded from 12 control and 7 Syt1-deficient cones evoked by a depolarizing step. The arrow points to transient $I_{Ca}$ inhibition in control cones. (**C**) $I_{Ca}$ traces from a control cone in response to a depolarizing step in control conditions, in the presence of 20 mM HEPES, and after washout of HEPES. (**D**) The paired-pulse protocol used to isolate $I_{inhibition}$ in **E–F**) and an example of $I_{Ca}$ recorded from a control cone (same cell as shown in **C**). (**E**) Average $I_{inhibition}$ obtained by subtracting $I_{Ca}$ during the second step ($I_{Ca(2)}$) from the first step ($I_{Ca(1)}$) of the paired-pulse protocol (subtraction traces were smoothed for clarity). Dashed lines indicate the duration of the depolarizing step. (**F**) Peak amplitude of $I_{inhibition}$ from control cones in control conditions, control cones with 20 mM HEPES, control cones after washout, and Cone$^{Syt1CKO}$ cones in control conditions. Inset image shows $I_{inhibition}$ amplitude measurement for the control condition. Control: $23.5 \pm 3.6$, $n = 5$; HEPES: $2.8 \pm 0.5$, $n = 5$; washout: $14.5 \pm 3.8$, $n = 4$; Cone$^{Syt1CKO}$: 0, $n = 4$. Control vs. HEPES: $p=0.04$, control vs. wash: $p=0.07$ (repeated measures one-way ANOVA); control vs. Cone$^{Syt1CKO}$: $p=0.02$ (Mann-Whitney test). One cone was lost before washout so it could not be included in ANOVA analysis. *$p \leq 0.04$.

DOI: https://doi.org/10.7554/eLife.45946.005

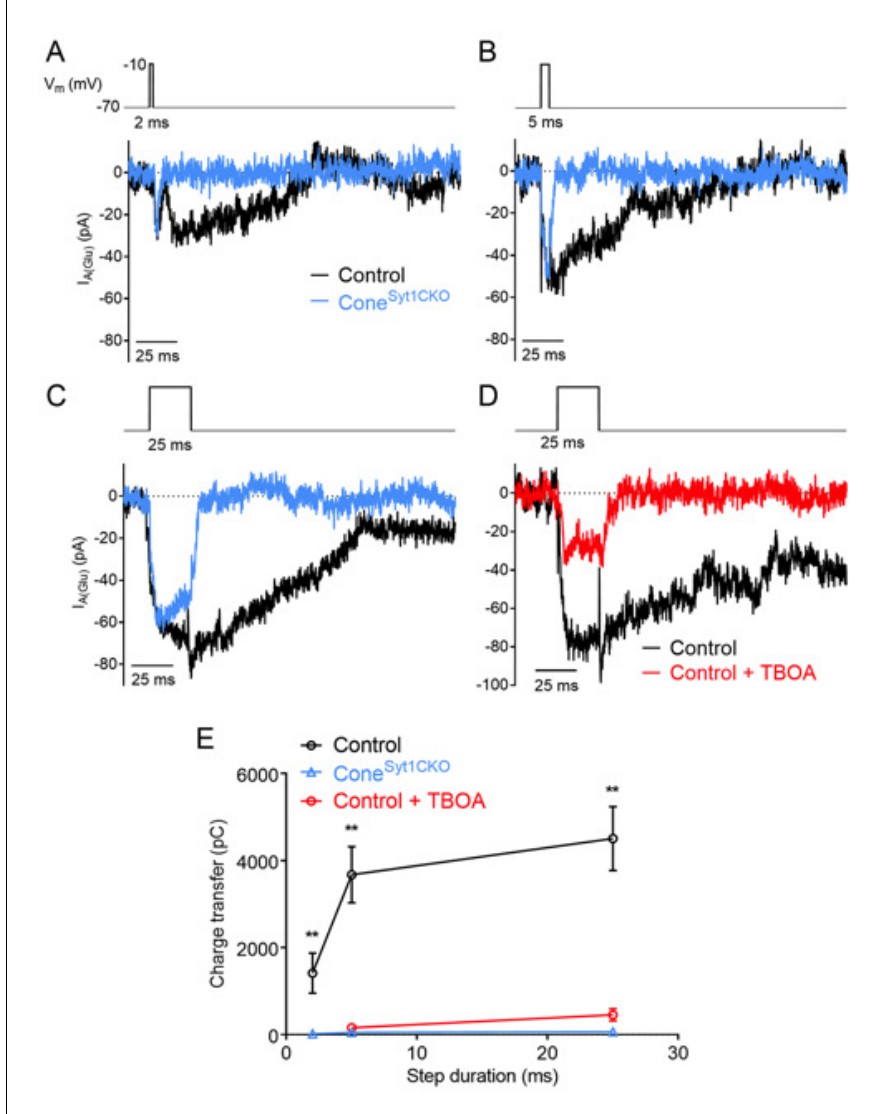

**Figure 5.** Glutamate release is eliminated in cones lacking Syt1. Example $I_{A(Glu)}$ recordings from control and Syt1-deficient cones in response to 2 (A), 5 (B), and 25 (C) ms depolarizing steps (−70 to −10 mV). (D) Example $I_{A(Glu)}$ recordings from a control cone evoked by 25 ms steps in control conditions and following bath application of the glutamate transport inhibitor, TBOA (100 µM). (E) $I_{A(Glu)}$ charge transfer as a function of step duration in control cones, Cone$^{Syt1CKO}$ cones, and control cones in the presence of 100 µM TBOA. Charge transfer was measured from the end of the test step. 2 ms: control: 1415 ± 463 pC, $n$ = 7 cones; Cone$^{Syt1CKO}$: 20.7 ± 7.6 pC, $n$ = 9 cones; control vs. Cone$^{Syt1CKO}$, p=0.0039. 5 ms: control: 3675 ± 646 pC, $n$ = 19 cones; Cone$^{Syt1CKO}$: 49.8 ± 11.3 pC, $n$ = 8 cones; control +TBOA: 160.3 ± 51.3 pC, $n$ = 7 cones; Cone$^{Syt1CKO}$ vs. control: p=0.0014, TBOA vs. control: p=0.0033. 25 ms: control: 4507 ± 734 pC, $n$ = 17 cones; Cone$^{Syt1CKO}$: 64.9 ± 29.9 pC, $n$ = 10 cones; control +TBOA: 455 ± 139 pC, $n$ = 9 cones; Cone$^{Syt1CKO}$ vs. control: p=0.0003, TBOA vs. control: p=0.0012 (t-tests corrected for multiple comparisons).

DOI: https://doi.org/10.7554/eLife.45946.006

evoked by the second step from $I_{Ca}$ evoked by the first step of the paired-pulse protocol, we could isolate and quantify the inhibitory component of the current ($I_{inhibition}$, *Figure 4E–F*). While the size of $I_{inhibition}$ averaged 23.5 pA in control cones, $I_{inhibiton}$ was undetectable in all cones that lacked Syt1 (*Figure 4F*), indicating that Syt1 is necessary for evoked exocytosis in cones.

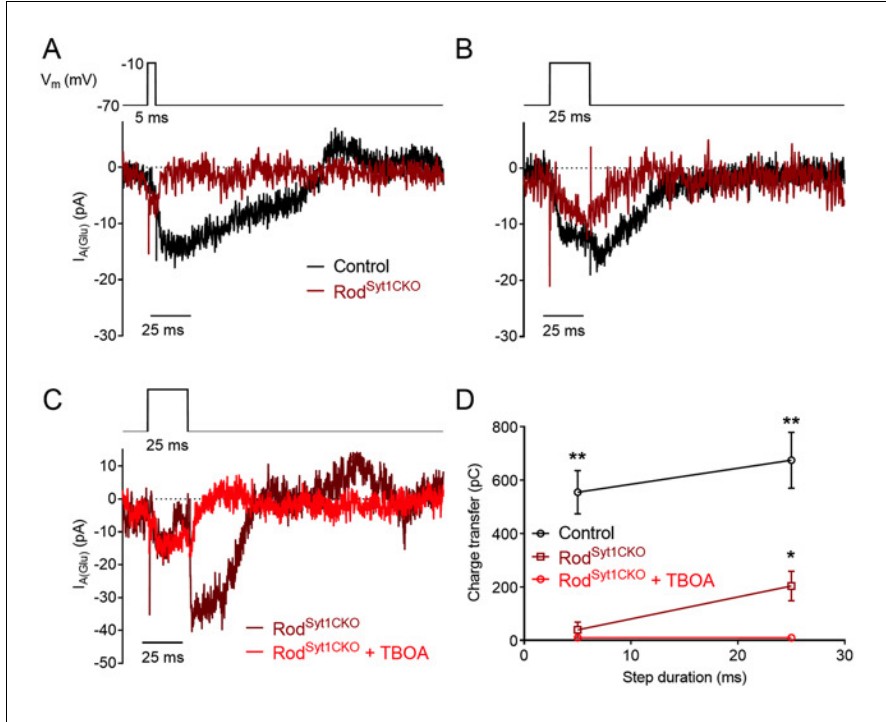

**Figure 6.** Glutamate release is diminished in rods lacking Syt1. Example $I_{A(Glu)}$ recordings from control and Syt1-deficient rods in response to 5 (**A**) and 25 (**B**) ms depolarizing steps ($-70$ to $-10$ mV). (**C**) Example $I_{A(Glu)}$ recordings from a Syt1-deficient rod evoked by 25 ms steps in control conditions and following bath application of the glutamate transport inhibitor, TBOA (300 μM). (**D**) $I_{A(Glu)}$ charge transfer as a function of step duration in control rods, Rod$^{Syt1CKO}$ rods, and in Rod$^{Syt1CKO}$ rods in the presence of 300 μM TBOA. 5 ms: control: $555 \pm 81$ pC, $n = 39$ rods; Rod$^{Syt1CKO}$: $39.1 \pm 29,3$ pC, $n = 31$ rods; Rod$^{Syt1CKO}$ + TBOA: $10.0 \pm 2.9$ pC, $n = 6$ rods; Rod$^{Syt1CKO}$ vs. control: p<0.0001. 25 ms: control: $674 \pm 105$ pC, $n = 37$ rods; Rod$^{Syt1CKO}$: $203.7 \pm 55.7$ pC, $n = 44$ rods; Rod$^{Syt1CKO}$ + TBOA: $9.6 \pm 7.5$ pC, $n = 8$ rods; Rod$^{Syt1CKO}$ vs. control: p<0.0001 (t-tests corrected for multiple comparisons), Rod$^{Syt1CKO}$ vs. Rod$^{Syt1CKO}$ + TBOA: p=0.0012 (Welch's t-test). \*\*p<0.0001, \*p=0.0012.
DOI: https://doi.org/10.7554/eLife.45946.007

## Glutamate release is reduced in Syt1-deficient photoreceptors

As a second technique to evaluate exocytosis in rods and cones, we utilized a retinal flatmount preparation to record presynaptic glutamate transporter-mediated anion tail currents ($I_{A(Glu)}$) evoked by depolarizing stimuli of varying duration (*Hasegawa et al., 2006*; *Szmajda and Devries, 2011*). Rod ribbons are surrounded by glutamate transporters EAAT2 and EAAT5 (*Arriza et al., 1997*; *Eliasof et al., 1998*; *Hasegawa et al., 2006*) while cones express two EAAT2 splice variants (*Eliasof et al., 1998*; *Rowan et al., 2010*; *Schneider et al., 2014*). Glutamate reuptake into the presynaptic terminal by any of these isoforms activates a robust uncoupled anion conductance (*Arriza et al., 1997*; *Schneider et al., 2014*) in rods and cones (*Grant and Werblin, 1996*; *Picaud et al., 1995*). In our experiments, Cl⁻ was replaced with the permeant anion isothiocyanate in the patch pipette solution to potentiate $I_{A(Glu)}$ (*Eliasof and Jahr, 1996*).

Control cones exhibited inward $I_{A(Glu)}$ tail currents in response to depolarizing steps to $-10$ mV of 2, 5, and 25 ms duration (*Figure 5A–C,E*). The glutamate transport inhibitor TBOA (100 μM) inhibited $I_{A(Glu)}$, demonstrating that these tail currents were generated by presynaptic glutamate transporter activity (*Figure 5D,E*). $I_{A(Glu)}$ charge transfer is linearly related to glutamate release (*Otis and Jahr, 1998*). *Figure 6E* plots charge transfer (measured from the end of the test step until the current returned to baseline) as a function of test step duration. Currents evoked in cones by longer 500 ms stimuli were not blocked by TBOA, suggesting that other currents (e.g., Ca²⁺-activated Cl⁻ currents) contributed to responses evoked by this stimulus (*Barnes and Hille, 1989*; *Cia et al., 2005*; *MacLeish and Nurse, 2007*; *Mercer et al., 2011*). Consistent with earlier results suggesting a

complete loss of release from cones, $I_{A(Glu)}$ was eliminated at all three stimulus durations in Syt1-deficient cones from Cone[Syt1CKO] retinas (**Figure 5**).

Control rods exhibited inward $I_{A(Glu)}$ tail currents in response to steps to −10 mV of 5 and 25 ms duration (**Figure 6**). In agreement with results from cones, $I_{A(Glu)}$ was eliminated in Syt1-deficient rods from Rod[Syt1CKO] retinas when stimulated with a 5 ms test step. However, unlike cones, some rods lacking Syt1 appeared capable of exocytosis that generated $I_{A(Glu)}$ when stimulated by 25 ms depolarizing steps (**Figure 6C–D**). To confirm that the tail currents measured from Syt1-deficient rods reflected $I_{A(Glu)}$, these experiments were repeated while TBOA (300 μM) was included in the bath solution. TBOA inhibited $I_{A(Glu)}$ in Rod[Syt1CKO] rods evoked by 5 and 25 ms steps (**Figure 6C–D**), indicating that these currents were due to activation of glutamate transporters. With 500 ms steps, TBOA blocked roughly half of the depolarization-evoked inward current, again suggesting that other currents contribute to responses evoked by this stimulus. Together, these results suggest that Syt1 mediates fast, phasic release evoked by brief elevations of $Ca^{2+}$ in both rods and cones. While Syt1 appears to mediate all evoked release in cones, other $Ca^{2+}$ sensors may be capable of stimulating release from rods given a sufficiently high concentration and/or long duration of $Ca^{2+}$ entry.

In the absence of stimulation, we observed spontaneous fast, monophasic inward currents in both rods and cones when they were voltage-clamped at −70 mV, below the activation threshold for $I_{Ca}$ (**Figure 7**). These events represent glutamate transporter anion currents activated by spontaneous fusion of vesicles in the recorded photoreceptor (**Cork et al., 2016**; **Grabner et al., 2015**; **Hasegawa et al., 2006**; **Szmajda and Devries, 2011**). Spontaneous $I_{A(Glu)}$ events were blocked by TBOA ($n$ = 9 rods, 8 cones). Unlike evoked release, and consistent with other synapses (**Schneggenburger and Rosenmund, 2015**), spontaneous release was not abolished in the absence of Syt1. Neither the quantal amplitude of spontaneous $I_{A(Glu)}$ events nor the frequency of spontaneous release differed between control and Cone[Syt1CKO] cones (**Figure 7A**). By contrast, the frequency of spontaneous release increased significantly in rods from Rod[Syt1CKO] retinas (**Figure 7B**), consistent with the suggested role for Syt1 in clamping spontaneous vesicle fusion that is mediated by a different $Ca^{2+}$ sensor (**Kaeser and Regehr, 2014**). The quantal amplitude did not differ between control rods and rods lacking Syt1 (**Figure 7B**).

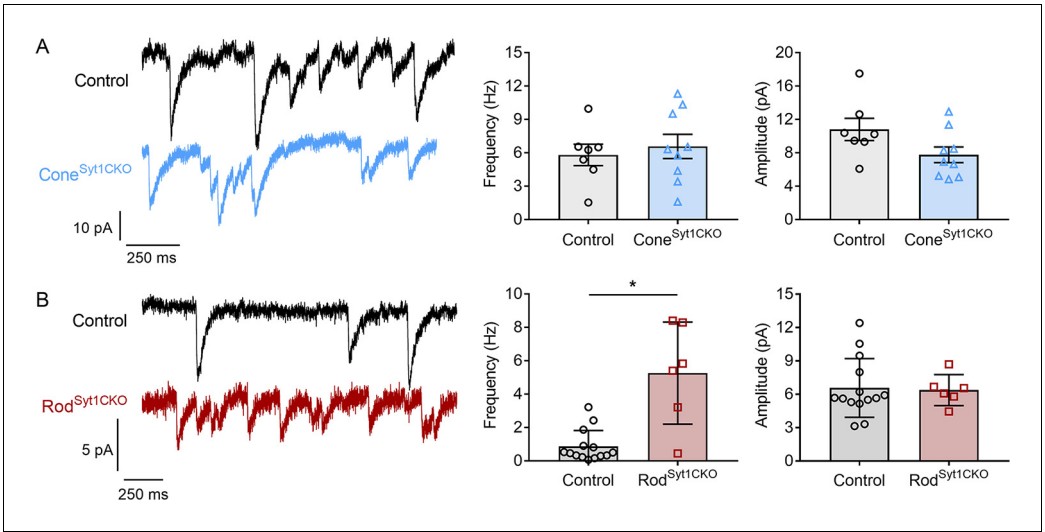

**Figure 7.** Spontaneous release persists in photoreceptors lacking Syt1. (**A**) Traces showing spontaneous $I_{A(Glu)}$ events from control and Cone[Syt1CKO] cones held at −70 mV. Frequency: control: 5.8 ± 0.96 Hz, $n$ = 7 cones; Cone[Syt1CKO]: 6.6 ± 1.09 Hz, $n$ = 9 cones; p=0.62 (t-test). Amplitude: control: 10.8 ± 1.34 pA, $n$ = 7 cones; Cone[Syt1CKO]: 7.8 ± 0.95 pA, $n$ = 9 cones; p=0.08 (t-test). (**B**) Traces showing spontaneous $I_{A(Glu)}$ events from control and Rod[Syt1CKO] rods held at −70 mV. Frequency: control: 0.9 ± 0.25 Hz, $n$ = 14 rods; Rod[Syt1CKO]: 5.3 ± 1.25 Hz, $n$ = 6 rods; p=0.004 (Mann-Whitney test). Amplitude: control: 6.6 ± 0.70 pA, $n$ = 14 rods; Rod[Syt1CKO]: 6.4 ± 0.57 pA, $n$ = 6 rods; p=0.87 (t-test). *p=0.004.
DOI: https://doi.org/10.7554/eLife.45946.008

## The absence of Syt1 does not reduce rod or cone $I_{Ca}$

The results described above strongly suggested that Syt1 is the chief mediator linking $Ca^{2+}$ influx to exocytosis in rods and cones. Another possible explanation for reduced neurotransmission by Syt1-deficient cells could be impaired $I_{Ca}$, although results above suggested this was not the case (*Figure 4*). We compared the amplitude and voltage dependence of $I_{Ca}$ in control and Syt1-deficient rods and cones. In cones, $I_{Ca}$ was measured using both a voltage ramp and depolarizing step series (*Figure 8A–C*). The current-voltage relationship obtained in cones with the voltage ramp (−90 to +60 mV, 0.5 mV/ms) closely matched the charge-voltage relationship ($Q_{Ca} = I_{Ca}$ integrated over the step duration) evoked by a series of 50 ms depolarizing voltage steps (*Figure 8A*). $I_{Ca}$ and $Q_{Ca}$ magnitudes were slightly larger in Cone$^{Syt1CKO}$ cones compared to control cones; this difference was statistically significant for $I_{Ca}$ but not $Q_{Ca}$ (*Figure 8D*). Voltage at half-maximum ($V_{0.5}$) values (obtained by fitting voltage-activation relationships with Boltzmann functions) of $I_{Ca}$ and $Q_{Ca}$ in Syt1-deficient cones did not differ significantly from controls but tended to be shifted slightly negative. Because $H^+$-mediated inhibition of $I_{Ca}$ reduces current amplitude and shifts voltage dependence positively (*Barnes et al., 1993*), these small differences in amplitude and $V_{0.5}$ likely arose because control cones were subject to $H^+$-mediated $I_{Ca}$ inhibition but Cone$^{Syt1CKO}$ cones were not (*Figure 4*). Furthermore, such differences would be expected to confer a small gain-of-function to Syt1-deficient cones and potentiate, rather than reduce, exocytosis. Unlike cones that have multiple ribbons,

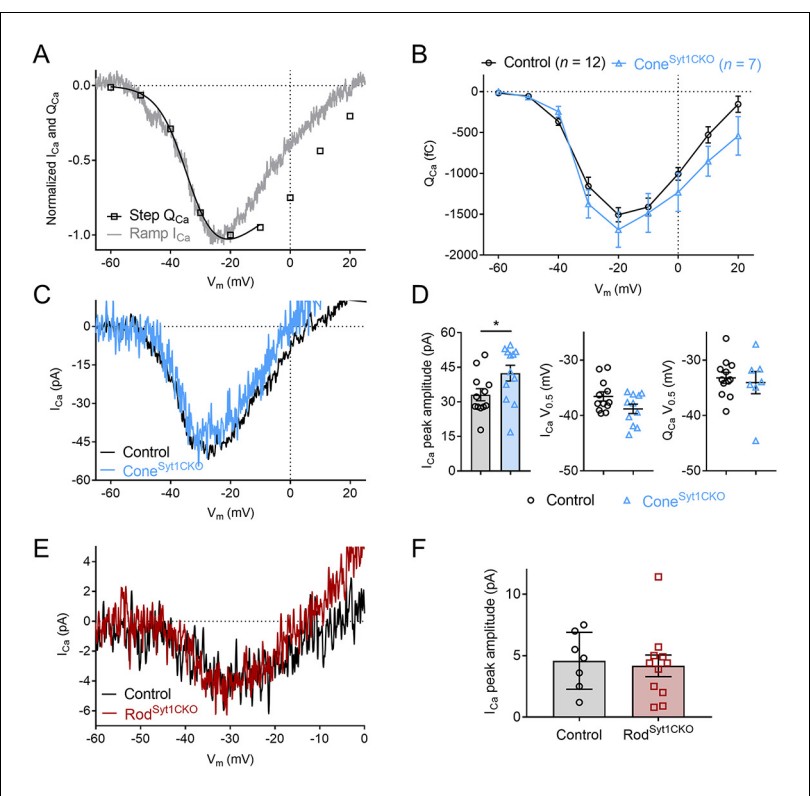

**Figure 8.** $I_{Ca}$ is not reduced in photoreceptors lacking Syt1. (A) Example current-voltage relationship of ramp-evoked $I_{Ca}$ and step-evoked $Q_{Ca}$ (a Boltzmann function adjusted for $Ca^{2+}$ driving force was fit to $Q_{Ca}$) in a control cone. (B) Average $Q_{Ca}$ as a function of step voltage for control and Cone$^{Syt1CKO}$ cones. No means differed significantly (t-tests corrected for multiple comparisons). (C) Example ramp-evoked $I_{Ca}$ traces from a control and Cone$^{Syt1CKO}$ cone. (D) Ramp-evoked $I_{Ca}$ peak amplitude (left) and $V_{0.5}$ values (middle), and step-evoked $Q_{Ca}$ $V_{0.5}$ values (right) from control and Cone$^{Syt1CKO}$ cones. $I_{Ca}$ amplitude: Control: 33.1 ± 2.63 pA, $n$ = 12; Cone$^{Syt1CKO}$: 42.5 ± 3.43 pA, $n$ = 12; p=0.04 (t-test). $I_{Ca}$ $V_{0.5}$: Control: −36.6 ± 0.83 mV, $n$ = 12; Cone$^{Syt1CKO}$: −38.8 ± 0.84 mV, $n$ = 11; p=0.07 (t-test). $Q_{Ca}$ $V_{0.5}$: Control: −33.2 ± 0.98 mV, $n$ = 12; Cone$^{Syt1CKO}$: −34.1 ± 2.01 mV, $n$ = 7; p=0.67 (t-test). (E) Average $I_{Ca}$ traces from control and Rod$^{Syt1CKO}$ rods. (F) Ramp-evoked $I_{Ca}$ peak amplitude from control and Rod$^{Syt1CKO}$ rods. Control: 4.6 ± 0.87 pA, $n$ = 7; Rod$^{Syt1CKO}$: 4.2 ± 0.88 pA, $n$ = 11; p=0.76 (t-test).

DOI: https://doi.org/10.7554/eLife.45946.009

mouse rods possess only a single ribbon and accordingly $I_{Ca}$ was much smaller in rods than cones. The amplitude of $I_{Ca}$ measured using voltage ramps did not differ between control and Syt1-deficient rods, and the two genotypes did not appear to differ in voltage dependence (*Figure 8E–F*). These results confirm that the marked neurotransmission defects in photoreceptors lacking Syt1 were not due to a reduction in presynaptic $Ca^{2+}$ influx.

## OPL architecture is maintained when rods or cones lack Syt1

Mutations of active zone proteins in the presynaptic terminals of photoreceptors—especially those associated with $Ca^{2+}$ channels—often lead to disordered synaptic structure, with malformed ribbons and defective photoreceptor wiring (*Bech-Hansen et al., 1998*; *Haeseleer et al., 2004*; *Kerov et al., 2018*; *Mansergh et al., 2005*; *Strom et al., 1998*; *Wang et al., 2017*). These mutated proteins not only disrupt vesicle release (e.g., due to altered $I_{Ca}$), but also play important roles in maintaining structural integrity of the ribbon or release sites. The role of synaptic communication per se in promoting and maintaining synaptic integrity remains unclear.

Photoreceptor synapses of 4–6 week old mice were examined using transmission electron microscopy (TEM) and immunohistochemistry (IHC). TEM images of the OPL from 5 week old control and Rod[Syt1CKO] retinas showed that rods of both genotypes contained well-formed synaptic ribbons anchored at the base of rod spherules with close apposition of the canonical triad of invaginating postsynaptic elements (presumably a central rod bipolar cell dendrite and horizontal cell dendrites on either side of the ribbon, *Figure 9A*). A dense cloud of vesicles, some of which were tethered to ribbons, surrounded rod ribbons. IHC experiments showed that control and Syt1-deficient rod ribbons were approximately equal in size and both exhibited the stereotypical horseshoe shape. Cone ribbons in Cone[Syt1CKO] retinas also appeared unchanged (*Figure 9B*). Consistent with the finding that $I_{Ca}$ was undiminished in the absence of Syt1, rods and cones lacking Syt1 maintained clusters of $Ca_V1.4$ channels below their ribbons (*Figure 9B*). Some studies have suggested the presence of the $Ca_V1.3$ $Ca^{2+}$ channels in photoreceptors (*Kersten et al., 2010*; *Morgans et al., 2005*; *Xiao et al., 2007*), but deletion of $Ca_V1.3$ in mice has only mild effects on the ERG a- and b-waves (*Busquet et al., 2010*; *Shi et al., 2017*; *Wu et al., 2007*), so we did not examine the distribution of $Ca_V1.3$. Measurements of the thickness of the ONL, OPL, and INL from IHC images did not differ among Rod[Syt1CKO], Cone[Syt1CKO], and their respective littermate control retinas (*Figure 9C*). Furthermore, the density of cone terminals in the OPL, assessed by counting tdTomato-positive cone terminals in flatmount retinas, did not differ between Cone[Syt1CKO] and control (*Figure 9D*).

Unlike mutations of postsynaptic proteins in ON bipolar cell dendrites (*Ball et al., 2003*; *Dhingra et al., 2000*; *Pinto et al., 2007*; *Ray et al., 2014*; *Tagawa et al., 1999*), mutations in critical presynaptic proteins at photoreceptor ribbon synapses typically cause horizontal and bipolar cells to extend dendrites beyond their normal site of termination in the OPL and form ectopic synapses in the ONL (*Cao et al., 2015*; *Haeseleer et al., 2004*; *Kerov et al., 2018*; *Mansergh et al., 2005*; *tom Dieck et al., 2005*; *tom Dieck et al., 2012*). We compared the anatomy of horizontal cell and bipolar cell dendrites in control, Rod[Syt1CKO], and Cone[Syt1CKO] retinas and found no evidence for sprouting of dendrites or an increase of ectopic synapses in the ONL. The diagram in *Figure 1C* illustrates the molecules used to identify different cell types. Triple labeling with antibodies to CtBP2 (to label ribbons), Syt1, and calbindin (to label horizontal cells, *Figure 10*) or secretagogin (to label cone bipolar cells, *Figure 11*); and double labeling of ribbons (with antibodies to CtBP2) and rod bipolar cells (with antibodies to PKCα, *Figure 12*) demonstrated that postsynaptic elements continued to make proper contacts with rod and cone ribbons in the OPL whether or not Syt1 was present in the associated presynaptic terminals. The ribbon protein Ribeye is an alternate transcript of a transcription factor, CtBP2. These proteins share a common B-domain, but Ribeye contains an additional A-domain. Antibodies to CtBP2 used to label Ribeye at synaptic ribbons therefore also label CtBP2 in the nucleus. We adjusted CtBP2 signal intensity to maximize visibility of the ribbons, causing some differences among preparations in nuclear staining; however, this variability did not appear to be systematically related to mouse phenotype. While we focused on mice 4–6 weeks of age, photoreceptor synapses of 10-week-old Rod[Syt1CKO] and Cone[Syt1CKO] mice also appeared unchanged (data not shown). These results revealed that both rod and cone synapses develop and are maintained normally in Rod[Syt1CKO] and Cone[Syt1CKO] mice, suggesting that neither Syt1-mediated activity nor Syt1 itself are required for establishing or maintaining photoreceptor ribbon structures or their contact with postsynaptic neurons, even weeks after synaptic maturation.

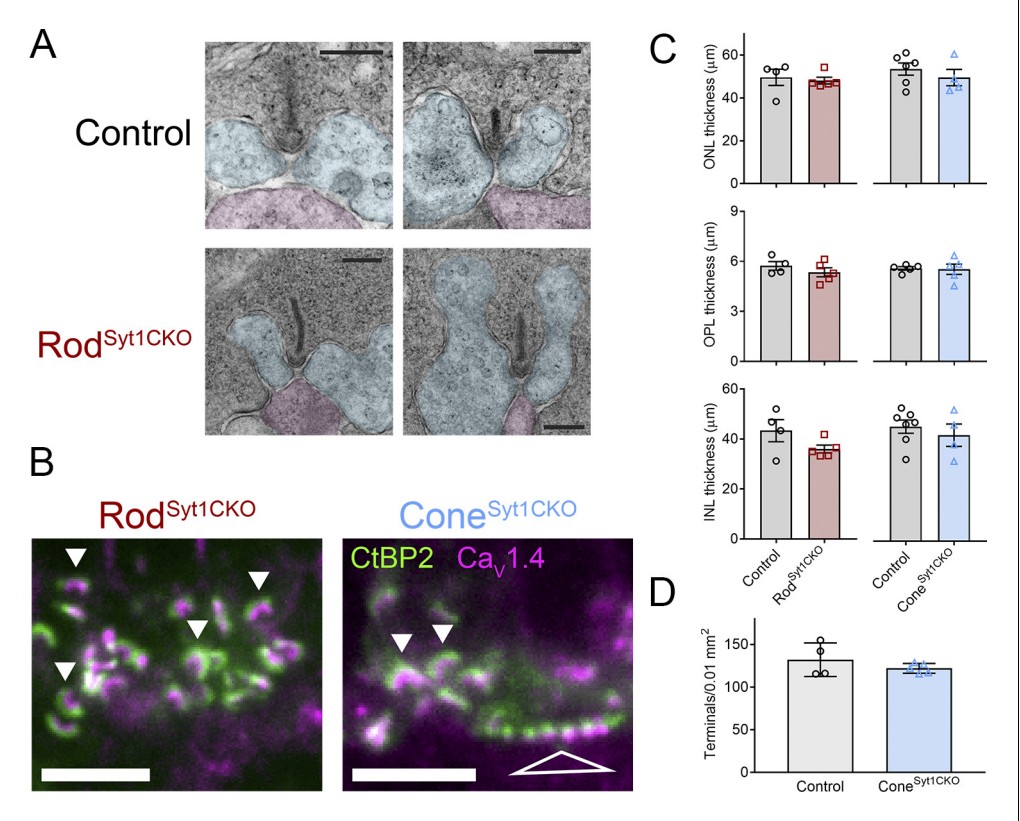

**Figure 9.** Photoreceptor synapses and retinal lamination appear unchanged in Syt1-mutant retinas. (**A**) TEM images of two rod synapses each from a control and Rod[Syt1CKO] retina with pseudocoloring of presumed horizontal cell (blue) and rod bipolar cell (magenta) dendrites. Scale bars = 200 nm. (**B**) Images centered on the OPL of Rod[Syt1CKO] and Cone[Syt1CKO] retinas co-labeled with antibodies to $Ca_V1.4$ (magenta) and CtBP2 (ribbons, green). Solid arrowheads indicate exemplar rod ribbon complexes and the open triangle points to numerous ribbon complexes within a cone terminal. Scale bars = 5 μm. (**C**) Measurements of ONL, OPL, and INL thickness from Rod[Syt1CKO], Cone[Syt1CKO], and their respective control retinas. ONL: rod control: 49.6 ± 3.79 μm, $n$ = 4; Rod[Syt1CKO]: 48.1 ± 1.56 μm, $n$ = 5; p=0.69; cone control: 53.5 ± 2.88 μm, $n$ = 6; Cone[Syt1CKO]: 49.5 ± 3.84 μm, $n$ = 4; p=0.42. OPL: rod control: 5.7 ± 0.26 μm, $n$ = 4; Rod[Syt1CKO]: 5.3 ± 0.27 μm, $n$ = 5; p=0.34; cone control: 5.6 ± 0.10 μm, $n$ = 5; Cone[Syt1CKO]: 5.5 ± 0.31 μm, $n$ = 5; p=0.88. INL: rod control: 43.4 ± 4.42 μm, $n$ = 4; Rod[Syt1CKO]: 36.0 ± 1.58 μm, $n$ = 5; p=0.13; cone control: 45.8 ± 2.64 μm, $n$ = 7; Cone[Syt1CKO]: 42.4 ± 4.49 μm, $n$ = 4; p=0.50 (t-tests). (**D**) Density of cone terminals per 0.01 mm² in control and Cone[Syt1CKO] retinas. Control: 132.2 ± 9.81, $n$ = 4 retinas; Cone[Syt1CKO]: 122.1 ± 2.59, $n$ = 5 retinas; p=0.30 (t-test).
DOI: https://doi.org/10.7554/eLife.45946.010

The following source data is available for figure 9:

**Source data 1.** Data for *Figure 9C*: Measurements of outer nuclear layer (ONL), outer plexiform layer (OPL, and inner nuclear layer (INL) thickness from Rod[Syt1CKO], Cone[Syt1CKO], and their respective control retinas.
DOI: https://doi.org/10.7554/eLife.45946.011

## Discussion

Our data demonstrate that Syt1 is the major $Ca^{2+}$ sensor linking presynaptic $Ca^{2+}$ and vesicular exocytosis at photoreceptor synaptic ribbon active zones. In both rods and cones, Syt1 appears to mediate fast, synchronous release evoked by brief depolarizing stimuli. Because the ERG b-wave reflects ON bipolar cell light responses, suppression of the b-wave without changes in the a-wave or photoreceptor $I_{Ca}$ in Rod[Syt1CKO] and Cone[Syt1CKO] retinas suggests that Syt1 also mediates continuous release in darkness in both rods and cones. The persistence of $I_{A(Glu)}$ evoked by 25 ms steps in rods lacking Syt1 suggests they may be capable of some slow, Syt1-independent evoked release. Spontaneous release also persisted in both rods and cones.

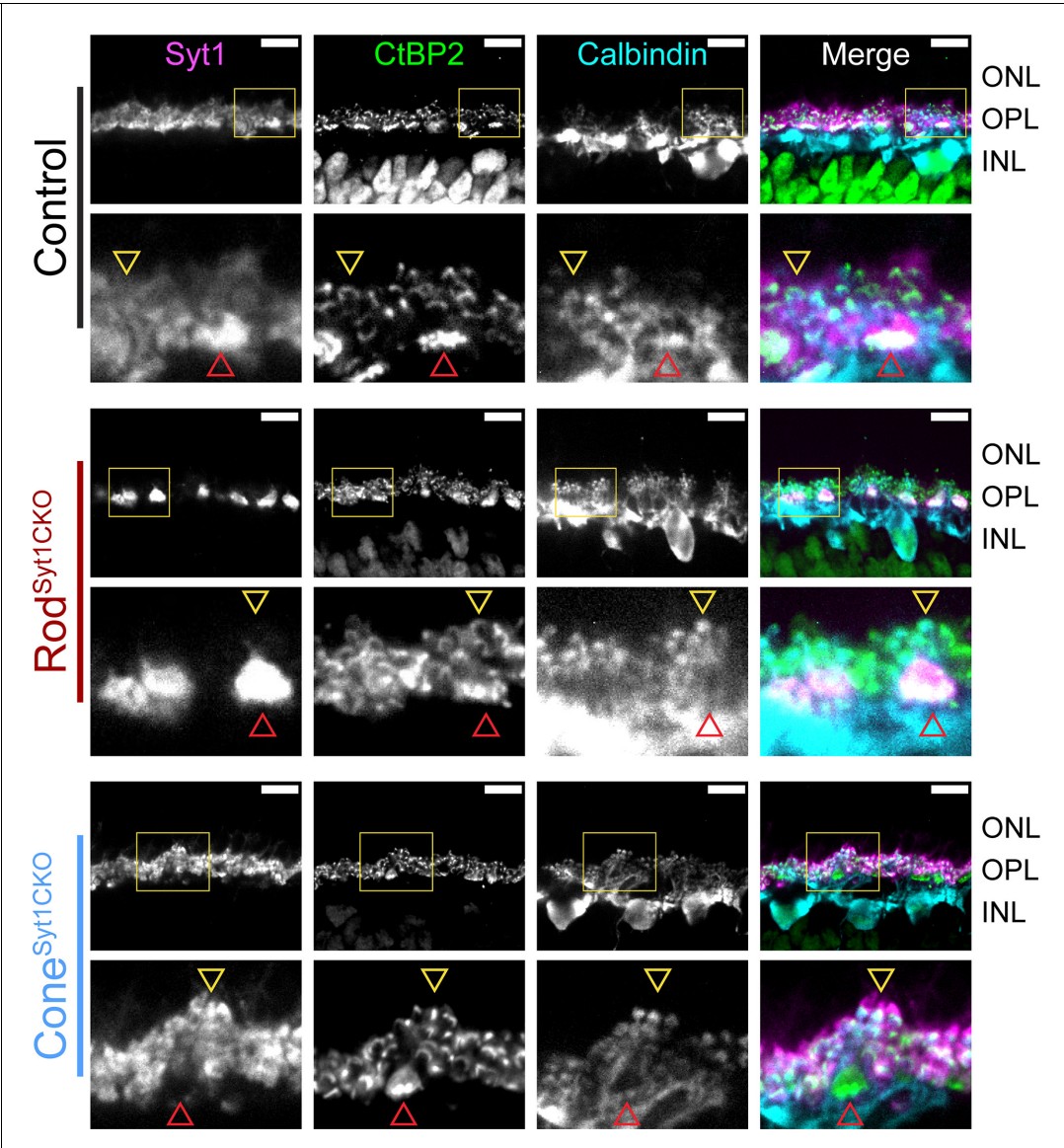

**Figure 10.** Horizontal cell dendrites maintain contact with Syt1-deficient rod and cone terminals in the OPL. Images of control, Rod[Syt1CKO], and Cone[Syt1CKO] retinas labeled with antibodies to Syt1 (magenta), CtBP2 (ribbons, green), and calbindin (horizontal cells, cyan). The top row of images for each genotype contain yellow boxes that indicate the boundaries of the high magnification images below. Red arrowheads point to representative cone terminals, yellow arrowheads point to representative rod terminals. Scale bars = 10 μm.
DOI: https://doi.org/10.7554/eLife.45946.012

## Other possible Ca²⁺ sensors in photoreceptors

Consistent with other synapses where Syt1 has been shown to inhibit spontaneous fusion of vesicles (*Kavalali, 2015*), the frequency of spontaneous miniature release events increased in rods lacking Syt1 (*Figure 7*). This result supports the hypothesis that Syt1 functions as a clamp of spontaneous fusion and that another Ca²⁺ sensor, perhaps Doc2, triggers spontaneous release (*Courtney et al., 2018*; *Groffen et al., 2010*). Syt1-deficient cones did not exhibit a similar increase in frequency suggesting that interactions between Syt1 and other exocytotic proteins differ in rods and cones.

Our results also suggest that a slower form of release, similar to asynchronous release described at other synapses, may persist in rods in the absence of Syt1. $I_{A(Glu)}$ was seen in rods lacking Syt1 from mice as old as P30, so it seems unlikely that these currents are due to the presence of residual Syt1 protein that had not been fully eliminated by Cre recombinase expression, although this

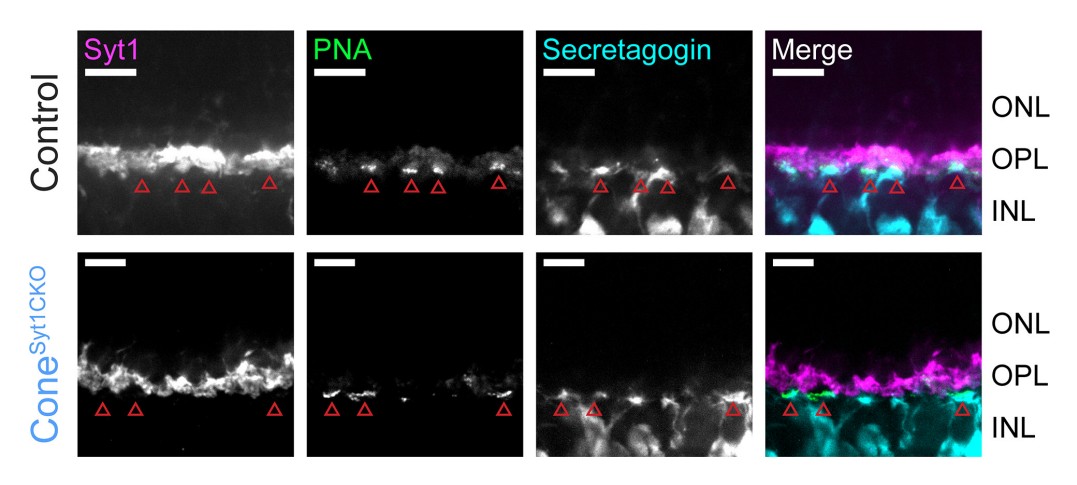

**Figure 11.** ON cone bipolar cell dendrites maintain contact with Syt1-deficient cone terminals in the OPL. Images of control and Cone[Syt1CKO] retinas labeled with PNA (green) to mark cone terminals and antibodies to Syt1 (magenta) and secretagogin (cone bipolar cells, cyan). Red arrowheads indicate cone terminals. Scale bars = 10 μm.

DOI: https://doi.org/10.7554/eLife.45946.013

remains a possibility. Alternatively, $Ca^{2+}$ sensors other than Syt1 may be able to trigger release given a sufficient $Ca^{2+}$ concentration and duration. Possible candidates include Doc2 and Syt7, which have both been suggested to mediate asynchronous release from neurons (*Bacaj et al., 2013*; *Luo et al., 2015*; *Turecek and Regehr, 2018*; *Yao et al., 2011*).

The exocytotic $Ca^{2+}$ sensor in amphibian photoreceptors shows a higher $Ca^{2+}$ affinity and lower $Ca^{2+}$ cooperativity ($n$ = 2–3) than most other synapses (*Duncan et al., 2010*; *Rabl et al., 2003*; *Rieke and Schwartz, 1996*). Unlike mammals, amphibian photoreceptors do not possess Syt1. Do these unusual properties arise from use of a different $Ca^{2+}$ sensor in amphibians or do other factors shape properties of release in mammalian and amphibian photoreceptors? Syt4 does not bind $Ca^{2+}$ ions but nevertheless imparts a linear $Ca^{2+}$ dependence to release from mature hair cells that involves the protein otoferlin with six $Ca^{2+}$-binding sites (*Cho et al., 2011*; *Johnson et al., 2010*).

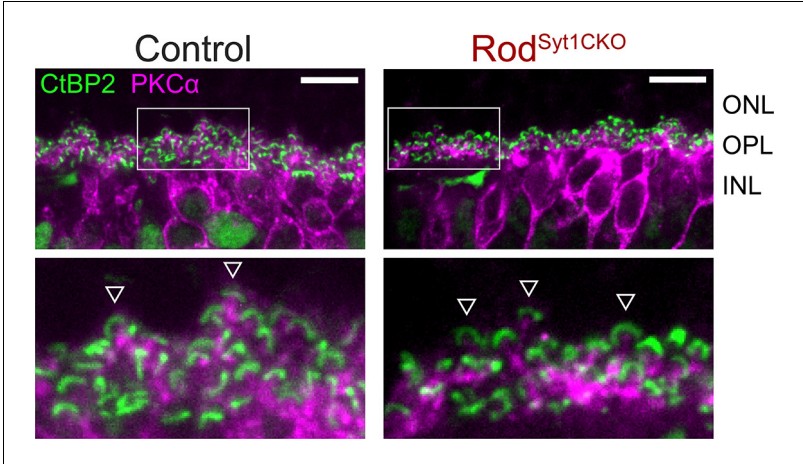

**Figure 12.** Rod bipolar cell dendrites maintain contact with rod ribbon synapses in the OPL of Rod[Syt1CKO] retinas. Images of control and Rod[Syt1CKO] retinas labeled with antibodies to CtBP2 (ribbons, green) and PKCα (rod bipolar cells, magenta). Boxes within images on the top indicate the boundaries of high magnification images shown below. Exemplar rod ribbon-rod bipolar cell dendrite contacts are indicated with arrowheads. Scale bars = 10 μm.

DOI: https://doi.org/10.7554/eLife.45946.014

Interactions between Syt1 and other SNARE complex members and intracellular signaling pathways have also been shown to influence cooperativity (*Sørensen et al., 2002*; *Stewart et al., 2000*; *Yang et al., 2005*).

## Implications of Syt1-mediated photoreceptor neurotransmission

Conventional and ribbon synapses both exhibit fast and slow components to exocytosis. At conventional synapses, these different kinetic components often involve the use of different $Ca^{2+}$ sensors, such as Syt1 or Syt2 for fast release and Syt7 or Doc2 for slow release (*Bacaj et al., 2013*; *Jarsky et al., 2010*; *Luo et al., 2015*). At ribbon synapses, kinetics of fast and slow release at bipolar and photoreceptor ribbon synapses are thought to be limited by the speed at which vesicles can be delivered to release sites at the base of the ribbon, which would not require different $Ca^{2+}$ sensors to mediate fast and slow release (*Bartoletti et al., 2010*; *Sterling and Matthews, 2005*). Consistent with this, release at mature ribbon synapses of cochlear hair cells appears to involve only a single sensor, otoferlin, and not Syt1, Syt2 or Syt7 (*Beurg et al., 2010*). On the other hand, like conventional synapses, Syt1 appears to mediate fast release from bipolar cell ribbon synapses while Syt7 contributes to slow asynchronous release (*Luo et al., 2015*). In cones, both phasic and tonic release are understood to occur exclusively at ribbons (*Jackman et al., 2009*; *Snellman et al., 2011*) and our results did not uncover any Syt1-independent release from cones. This suggests that Syt1-mediated release at ribbons, with kinetics dictated by vesicle delivery, is the dominant mechanism of both fast and slow exocytosis in mouse cones.

Fast release from rods uses ribbon-mediated mechanisms similar to cones (*Li et al., 2010*; *Van Hook and Thoreson, 2015*). However, unlike cones, rods are also capable of slow non-ribbon release stimulated by $Ca^{2+}$-induced $Ca^{2+}$ release (CICR) from intracellular ER stores (*Babai et al., 2010*; *Cadetti et al., 2006*; *Chen et al., 2014*; *Chen et al., 2013*; *Krizaj et al., 2003*; *Suryanarayanan and Slaughter, 2006*). Inhibition of CICR significantly reduced light responses of horizontal cells and bipolar cells in amphibian and mouse retina, suggesting that non-ribbon release contributes significantly to continuous release from rods in darkness (*Babai et al., 2010*; *Cadetti et al., 2006*; *Suryanarayanan and Slaughter, 2006*). We found that deletion of Syt1 eliminated most of the rod-driven b-wave and that only a small component of slow release persisted in rods lacking Syt1. This suggests that if, like bipolar cells, slow asynchronous release from rods employs a sensor other than Syt1, it is not likely to be a major contributor to continuous release in darkness. However, it is also possible that the increase in spontaneous release frequency in rods lacking Syt1 (*Figure 7*) might be large enough to obscure light-evoked changes in release mediated by another sensor and thereby block the b-wave.

Exocytosis mediated by the fast sensor Syt1 could help to ensure that vesicles that reach release sites are released with minimal delay. While fast sensors like Syt1 can promote fusion within a few hundred microseconds after $Ca^{2+}$ elevation, slower sensors generally take much longer (*Kaeser and Regehr, 2014*). The use of Syt1 may promote a close temporal association between the formation of $Ca^{2+}$ nanodomains and vesicle release (*Bartoletti et al., 2011*; *Van Hook and Thoreson, 2015*), even during continuous release in darkness.

## Maintained integrity of photoreceptor synapses lacking Syt1

Syt1-deficient photoreceptors exhibit undiminished $I_{Ca}$ and maintain synaptic contacts with postsynaptic horizontal and bipolar cells despite diminished synaptic communication. Immunohistochemistry and TEM data showed that photoreceptor synapses were morphologically indistinguishable in control, Rod$^{Syt1CKO}$, and Cone$^{Syt1CKO}$ retinas (*Figures 9–12*). Previous studies of mouse photoreceptors examining mutations or the absence of critical ribbon-associated proteins CaBP4 (*Haeseleer et al., 2004*), CAST (tom *tom Dieck et al., 2012*), Bassoon (*Specht et al., 2007*), and the $Ca_V1.4$ channel subunits $\alpha_1$ (*Chang et al., 2006*; *Mansergh et al., 2005*), $\beta2$ (*Ball et al., 2002*; *Katiyar et al., 2015*), and $\alpha_2\delta_4$ (*Wang et al., 2017*; *Wycisk et al., 2006*) have all demonstrated reduced synaptic contacts, malformed ribbons, and/or ectopic photoreceptor synapses with postsynaptic dendritic sprouting into the ONL, with concomitant synaptic transmission defects. It has also been suggested that photoreceptor synaptic dysfunction is the initiating event that causes horizontal cells to sprout ectopic dendrites in retinal degeneration (*Soto and Kerschensteiner, 2015*). That the absence of Syt1 from photoreceptors did not affect the development or maintenance (up to 10 postnatal weeks) of rod or

cone synapses was surprising given its integral role in photoreceptor neurotransmission. However, consistent with our results, *Cao et al. (2015)* found that expressing tetanus toxin to silence mouse photoreceptors throughout development caused only subtle morphological changes to photoreceptor ribbons and occasional ectopic sprouting of postsynaptic dendrites without major changes in ONL/OPL architecture. When the integral ribbon protein RIBEYE was deleted in mice, neurotransmission was diminished but OPL anatomy was not markedly altered (*Maxeiner et al., 2016*). Another study that deprived developing mice of visual experience found only a subtle effect on the formation of cone synapses, but not rod synapses (*Dunn et al., 2013*). Finally, mutations of postsynaptic mGluR6 signaling cascade members disrupt communication, but not synaptic anatomy, between photoreceptors and ON bipolar cells (*McCall and Gregg, 2008*). These results support the hypothesis that ribbon-associated $Ca^{2+}$ channel complexes, ribbon structural components, trans-synaptic protein interactions, or other $Ca^{2+}$-dependent processes (i.e., as a second messenger) may be the dominant factors establishing photoreceptor synaptic connectivity (*Joiner and Lee, 2015*; *Schmitz, 2009*). Because Syt1 is a vesicular protein with no direct association with the ribbon complex, our results are consistent with previous studies emphasizing the importance of proteins involved in active zone organization while suggesting a limited role for light-evoked neurotransmission in maintaining photoreceptor synapse integrity.

Spontaneous activity is understood to play a key role in synaptogenesis within the retina and throughout the CNS (*Andreae and Burrone, 2018*; *Soto et al., 2012*), and $Ca^{2+}$-dependent neurotransmission before eye opening has been implicated in cone-horizontal cell synapse formation specifically (*Raven et al., 2008*). Rods and cones lacking Syt1 remained capable of spontaneous release (*Figure 7*), so it is possible that Syt1-independent release is sufficient for both the formation and maintenance of photoreceptor synapses. It is also possible that Syt1 expression may rise briefly before its Cre-dependent deletion and this may be sufficient to establish photoreceptor wiring. Cre expression in both HRGP-*Cre* and *Rho-iCre* mice (*Le et al., 2004*; *Li et al., 2005*) rises in parallel with formation of photoreceptor synapses during the second postnatal week in mouse retina (*Blanks et al., 1974*). It is also possible that subtle morphological changes occurred that evaded our detection or that Syt1-deficient photoreceptor synapses could form properly but degenerate at time points beyond those evaluated by our study.

# Materials and methods

## Key resources table

| Reagent type (species) or resource | Designation | Source or reference | Identifiers | Additional information |
|---|---|---|---|---|
| Genetic reagent (mouse) | *HRGP-Cre* | PMID: 15635292 | NA | |
| Genetic reagent (mouse) | *Syt1^{flox}* | PMID: 28511701 | NA | *Syt1*: MGI:99667 |
| Genetic reagent (mouse) | Ai14 | Jackson Laboratories | RRID:IMSR_JAX:007914 | PMID: 20023653 |
| Genetic reagent (mouse) | *Rho-iCre* | Jackson Laboratories | RRID:IMSR_JAX:015850 | PMID: 15682388 |
| Antibody | Rabbit anti-calbindin | Swant | CB38; RRID:AB_2721225 | 1:10,000 |
| Antibody | Mouse anti-CtBP2 | BD | 612044; RRID:AB_399431 | 1:1000 |
| Antibody | Goat anti-CtBP2 | Santa Cruz | sc-5966; RRID:AB_2086774 | 1:250 |
| Antibody | Rabbit anti-Ca$_V$1.4 | Dr. Amy Lee, U. of Iowa | RRID:AB_2650487 | 1:1000 |
| Antibody | Peanut agglutinin (PNA, FITC conjugated) | Bionexus | BN-F44 | 1:100 |
| Antibody | Peanut agglutinin (PNA, Cy-5 conjugated) | Vector Laboratories | CL-1075 | 1:100 |

*Continued on next page*

*Continued*

| Reagent type (species) or resource | Designation | Source or reference | Identifiers | Additional information |
|---|---|---|---|---|
| Antibody | Rabbit anti-PKC | Santa Cruz | sc-10800; RRID:AB_2168560 | 1:200 |
| Antibody | Rabbit anti-PSD95 | Abcam | ab18258; RRID:AB_444362 | 1:500 |
| Antibody | Rabbit anti-secretagogin | Biovendor | RD181120100; RRID:AB_2034060 | 1:1000 |
| Antibody | Mouse anti-Syt1*Oyster550 | Synaptic Systems | 105 011C3; RRID:AB_887827 | 1:1000 |
| Antibody | Donkey anti-goat secondary (Alexa Fluor 488 conjugated) | Thermo Fisher Scientific | A11055; RRID:AB_142672 | 1:200 |
| Antibody | Goat anti-mouse-FITC | BD | 554001; RRID:AB_395197 | 1:200 |
| Antibody | Goat anti-rabbit secondaries (Alexa Fluor 488 and 568 conjugated) | Thermo Fisher Scientific | A11008, A11011; RRID:AB_143165, AB_143157 | 1:200 |
| Antibody | Donkey anti-rabbit secondary (Alexa Fluor 568 conjugated) | Thermo Fisher Scientific | A10042; RRID:AB_2534017 | 1:200 |
| Antibody | Donkey anti-rabbit secondary (Alexa Fluor 647 conjugated) | Abcam | Ab150075; RRID:AB_2752244 | 1:200 |

## Mice

Control and mutant mice were bred on predominantly or wholly C57/Bl6J backgrounds. Mice were kept on 12 hr dark-light cycles. HRGP-*Cre* and *Syt1*$^{flox}$ (Syt1: MGI:99667) mice have been described previously (*Le et al., 2004*; *Quadros et al., 2017*). *Rho-iCre* (RRID:IMSR_JAX:015850) and td-Tomato Cre reporter Ai14 (RRID:IMSR_JAX:007914) mice were obtained from Jackson Labs (*Li et al., 2005*; *Madisen et al., 2010*). Control mice for ERGs and rod single-cell recordings were *Rho-iCre*$^{negative}$, *Syt1*$^{flox/flox}$ mice. Control mice for cone single-cell recording experiments were HRGP-*Cre*$^{+}$, *Syt1*$^{+/flox}$, *tdTomato*$^{+}$ mice. Mice aged 3–6 weeks of both sexes were used. Euthanasia was conducted in accordance with AVMA Guidelines for the Euthanasia of Animals by $CO_2$ asphyxiation followed by cervical dislocation. All animal care and handling protocols were approved by the University of Nebraska Medical Center Institutional Animal Care and Use Committee.

## Electroretinography (ERG)

Ex vivo ERGs were recorded using an enucleated eyecup preparation and chamber as described in detail by *Newman and Bartosch (1999)*. Mice were dark-adapted for >30 min and then euthanized and dissected in darkness under infrared illumination. Eyes were enucleated with Graefe forceps and placed in Ames' solution bubbled with 95% $O_2$/5% $CO_2$. The anterior portion of the eye and the lens were removed while bathed in Ames' solution and discarded. Relieving cuts were made in the back half of the eye which was then everted over a Ag/AgCl pellet embedded in a small mound of dental wax in the bottom half of the recording chamber. The top half of the recording chamber with another Ag/AgCl electrode was then placed carefully over the eye and secured as described (*Newman and Bartosch, 1999*). The chamber was kept in darkness while it was transferred to the microscope stage. The eyecup was superfused with ~35°C Ames' solution bubbled with 95% $O_2$/5% $CO_2$ supplemented with $BaCl_2$ and glutamic acid (both 100 µM) for the duration of the experiment. Once in the recording configuration, the eyecup preparation was dark-adapted >20 min before light stimuli were delivered.

Light stimuli were generated by a 50 W halogen lamp focused onto a fiber optic with neutral density filters in the light path to attenuate intensity as needed. The light was directed onto the eyecup from above. The unattenuated light intensity at 580 nm was measured as $5.8 \times 10^5$ photons s$^{-1}$ µm$^{-2}$ using a laser power meter (Metrologic, Blackwood, NJ). In control mice, ERG b-wave

responses to flashes of unattenuated white light were 1.51 ± 0.24 (*n* = 4) times larger than responses to 580 nm light; the effective intensity of the unattenuated white light stimulus was therefore ~$8.8 \times 10^5$ photons $s^{-1}$ $\mu m^{-2}$, equivalent to approximately $7 \times 10^5$ photons $s^{-1}$ $rod^{-1}$ (*Lyubarsky et al., 2004*; *Naarendorp et al., 2010*).

For most experiments, single flash ERGs were evoked using a 20 ms flash. At low light intensity ($<10^3$ below maximum), stimulus trials were separated by at least 10 s. At higher light intensities, trials were separated by minutes. For flicker experiments, flashes of 20 ms duration with an equal duration of darkness between flashes (on-off square wave) were delivered. In flicker trials, responses to 25 consecutive square wave periods were recorded and the average ERG response was calculated by averaging the ERG from periods 5–25 to avoid contamination from the larger initial light-evoked waveform. These average flicker responses were normalized by dividing by the amplitude of the b-wave evoked by the first flash applied during the trial and corrected for non-zero slope (arising from slow ERG components or drift) to facilitate comparison among samples. ERGs were recorded using PClamp software (Axon Instruments/Molecular Devices; RRID:SCR_011323) in current clamp configuration, AC coupled at 0.1 Hz, and lowpass filtered at 3 kHz. A-wave amplitudes were measured from the pre-stimulus baseline and b-wave amplitudes were measured from the a-wave trough to the b-wave peak.

## Electrophysiology

Slice and flatmount experiments were performed on an upright fixed-stage microscope (Nikon E600FN) under a water-immersion objective (60×, 1.0 NA). Cell bodies were identified morphologically for rods or using tdTomato fluorescence for cones. Recording electrodes were positioned with Huxley-Wall micromanipulators (Sutter Instruments). Rods and cones were recorded in flatmount retinal preparations and cones were also recorded using vertical retinal slices prepared similarly to salamander slices (*Van Hook and Thoreson, 2013*). After obtaining a gigaohm seal, the patch was ruptured with gentle suction. Photoreceptor recordings were performed in voltage clamp using an Axopatch 200B (Axon Instruments, Molecular Devices) amplifier. Cone membrane currents from the Axopatch were filtered at 2 kHz. Some membrane currents were low-pass filtered post-hoc at 600 Hz to facilitate data presentation. Signals were digitized with a Digidata 1322A (Molecular Devices) and acquired with pClamp 10 software (Molecular Devices). Passive membrane resistance was subtracted from $I_{Ca}$ and $I_{A(Glu)}$ currents using P/8 subtraction. Voltages were not corrected for liquid junction potentials (CsGluconate pipette solution: 12.3 mV, KSCN pipette solution: 3.9 mV). For $V_{0.5}$ calculations, $I_{Ca}$ and $Q_{Ca}$ measurements were fit with Boltzmann functions adjusted for $Ca^{2+}$ driving force assuming a $Ca^{2+}$ reversal potential of +50 mV. $I_{A(Glu)}$ charge transfer was measured from the end of the test step until the current returned to baseline.

Whole cell recordings were performed at room temperature. Preparations were constantly superfused at ~1 mL/min with Ames solution (US Biological) bubbled with 95% $O_2$/5% $CO_2$. The solution in *Figure 5* in which 20 mM HEPES was supplemented to the Ames solution was prepared to maintain consistent [$Ca^{2+}$] and osmolarity by first dividing the control Ames medium into two portions. 20 mM HEPES was then added to one portion, its pH was readjusted to 7.3 with NaOH, and the solution was diluted with $H_2O$ to reach an osmolarity of 275–280 (approximately 10% dilution). The control solution was then also diluted 10% and its osmolarity was returned to 275–280 with NaCl. The intracellular pipette solution for $I_{Ca}$ measurements contained (in mM): 120 CsGluconate, 10 TEACl, 10 HEPES, 2 EGTA, 1 CaCl$_2$, 1 MgCl$_2$, 0.5 NaGTP, 5 MgATP, five phosphocreatine, pH 7.2–7.3. For $I_{A(Glu)}$ measurements, KSCN replaced CsGluconate in the intracellular solution and EGTA was raised to 5 mM. All chemical reagents were obtained from Sigma-Aldrich unless indicated otherwise. Membrane capacitance, membrane resistance, and access resistance values for cones in slices using the CsGluconate pipette solution averaged 9.4 ± 0.4 pF, 1.1 ± 0.07 GΩ, and 74 ± 4.0 MΩ (*n* = 30); and for cones in flatmounts using the KSCN solution were 6.2 ± 0.2 pF, 0.53 ± 0.05 GΩ, and 77 ± 6.5 MΩ (*n* = 27). For rods using the CsGluconate pipette solution these values averaged 3.3 ± 0.2 pF, 2.0 ± 0.4 GΩ, and 55 ± 12 MΩ (*n* = 8); and for the KSCN solution were 3.6 pF ± 0.2, 1.9 ± 0.2 GΩ, and 56 ± 7.9 MΩ (*n* = 12, all rods were recorded in flatmounts).

## Immunohistochemistry

Mice aged 4–6 weeks were euthanized as described and eyes were enucleated in Ames solution. For immunohistochemistry sections, the anterior portion of the eye was removed and the posterior eye-cup was fixed in 4% paraformaldehyde for 30–40 min, washed 6 times for 10 min each in PBS, and cryoprotected overnight at 4°C in 30% sucrose. Eyecups were embedded in OCT compound (Sakura Finetek USA) and stored at −80°C until sectioning at 20–30 µm with a cryostat (Leica CM 1800). For experiments in which tdTomato$^+$ cone terminal density was quantified, retinas were removed from the eyecup, fixed for 15 min in 4% paraformaldehyde, washed in PBS, and flatmounted photoreceptor side up. For antibody staining, retinal sections were treated with a blocking solution of 1% Triton X-100% and 6% donkey serum (Jackson Labs) before both the primary and secondary antibody application steps. Primary and secondary antibodies were diluted to working concentrations in blocking solution. Sections were incubated in primary antibodies at 4°C for 1–3 nights and in secondary antibodies at room temperature for 2–3 hr. Both retinal sections and flatmount retinas were mounted with Vectashield (Vector Labs, RRID:AB_2336787) before imaging. The specificities of anatomical markers have been described previously: calbindin (*Massey and Mills, 1996*), CtBP2 (*Schmitz et al., 2000*), PKCα (*Greferath et al., 1990*), PNA (*Blanks and Johnson, 1984*), PSD95 (*Koulen et al., 1998*), and secretagogin (*Puthussery et al., 2010*). Primary and secondary antibodies plus dilutions are listed in the Key Resources table. Confocal imaging was performed using NIS Elements software (Nikon, RRID:SCR_014329) and a laser confocal scanhead (Perkin Elmer Ultraview LCI) equipped with a cooled CCD camera (Hamamatsu Orca ER, RRID:SCR_017105) mounted on a Nikon E600FN microscope. Fluorescent excitation was delivered from an argon/krypton laser at 488, 568, or 648 nm wavelengths and emission was collected at 525, 607, and 700 nm, respectively. Filters were controlled using a Sutter Lambda 10–2 filter wheel and controller. The objective was controlled using a E662 z-axis controller (Physik Instrumente). Images were analyzed and adjusted using Nikon Elements, Fiji, and Adobe Photoshop software. TdTomato$^+$ cone terminal densities were quantified by averaging the density of multiple separate 0.01 mm$^2$ areas of random eccentricity per retina. Retinal layer thickness measurements were made using clear boundaries of layer-specific antibodies (e.g., Syt1 in control retinas clearly defines the boundaries of the OPL and IPL) measured every 10 µm on two different confocal planes (separated by >2 µm) from each cryosection, producing average thicknesses calculated from 20 to 40 measurements per mouse.

## Electron microscopy

Retinal pieces were fixed overnight at 4 deg C in 2% glutaraldehyde, 2% paraformaldehyde, and 0.1 M Sorensen's phosphate buffer (pH 7.4). After fixation, retinas were washed three times in phosphate-buffered saline and then placed in 1% osmium tetroxide. Samples were dehydrated through a graded ethanol series with each concentration (50%, 70%, 90%, 95%, 100%) applied for 3 min. Retinas were then washed three times with 100% propylene oxide. Samples were left overnight in a 1:1 mixture of Araldite embedding medium and propylene oxide, embedded in fresh Araldite in silicon rubber molds, and then placed in an oven at 65°C overnight. Resulting blocks were thin sectioned on a Leica UC6 ultramicrotome and placed on 200 mesh copper grids. Sections were stained with 1% uranyl acetate and Reynold's lead citrate. Sections were examined in a FEI Tecnai G2 TEM operated at 80 kV.

## Statistical analysis

Statistical analysis and data visualization were done using GraphPad Prism. Where applicable, p values were adjusted for multiple comparisons using the Holm-Šídák method for multiple one-way ANOVA and t-tests. Post-hoc testing for one-way ANOVAs was done using Dunnett's tests. The criterion for statistical significance was set at α = 0.05. Data is presented as mean ± SEM.

## Acknowledgements

The authors thank the UNMC Mouse Genome Engineering and Electron Microscopy core facilities for experimental assistance, Dr. Yun Le (Univ. of Oklahoma Health Sciences Center) for HRGP-*Cre* mice, Dr. Amy Lee (Univ. of Iowa) for supplying antibodies to Ca$_V$1.4, and Yumeng Shi for experimental assistance.

# Additional information

## Funding

| Funder | Grant reference number | Author |
|---|---|---|
| National Eye Institute | EY10542 | Wallace B Thoreson |
| Research to Prevent Blindness | Senior Scientific Investigator | Wallace B Thoreson |
| National Eye Institute | EY28848 | Justin J Grassmeyer |

The funders had no role in study design, data collection and interpretation, or the decision to submit the work for publication.

## Author contributions

Justin J Grassmeyer, Data curation, Formal analysis, Funding acquisition, Investigation, Methodology, Writing—original draft, Writing—review and editing; Asia L Cahill, Cassandra L Hays, Formal analysis, Investigation, Writing—review and editing; Cody Barta, Resources, Validation, Writing—review and editing; Rolen M Quadros, Resources, Validation; Channabasavaiah B Gurumurthy, Resources, Supervision, Validation, Writing—review and editing; Wallace B Thoreson, Conceptualization, Data curation, Formal analysis, Supervision, Funding acquisition, Investigation, Methodology, Project administration, Writing—review and editing

## Author ORCIDs

Justin J Grassmeyer (iD) https://orcid.org/0000-0001-9624-9027
Rolen M Quadros (iD) http://orcid.org/0000-0002-9798-9622
Channabasavaiah B Gurumurthy (iD) http://orcid.org/0000-0002-8022-4033
Wallace B Thoreson (iD) https://orcid.org/0000-0001-7104-042X

## Ethics

Animal experimentation: All animal care and handling protocols were approved by the University of Nebraska Medical Center Institutional Animal Care and Use Committee (protocols 15-027-00 and 15-028-04).

## Decision letter and Author response

Decision letter https://doi.org/10.7554/eLife.45946.017
Author response https://doi.org/10.7554/eLife.45946.018

# Additional files

## Supplementary files

• Transparent reporting form
DOI: https://doi.org/10.7554/eLife.45946.015

## Data availability

All data generated or analysed during this study are included in the manuscript and supporting files. Source data files have been provided for Figure 9.

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
