## [Decision Letter]

Thank you for submitting your article "Ca^2+^ sensor synaptotagmin-1 mediates exocytosis in mammalian photoreceptors" for consideration by *eLife*. Your article has been reviewed by three peer reviewers, and the evaluation has been overseen by a Reviewing Editor and Richard Aldrich as the Senior Editor. The following individual involved in review of your submission has agreed to reveal their identity: Henrique von Gersdorff (Reviewer #1).

The reviewers have discussed the reviews with one another and the Reviewing Editor has drafted this decision to help you prepare a revised submission.

Summary:

Grassmeyer and colleagues investigate the role of the calcium sensor synaptotagmin-1 (Syt1) for transmitter release from mouse cone and rod photoreceptors. The authors use a floxed Syt1 allele, which is crossed with either cone or rod – specific Cre driver lines to produce cell-specific Syt1 KOs. Glutamate release at rods and cones is then monitored by complementary approaches including electroretinography in a retinal in-vitro preparation, measurement of the proton inhibition of calcium currents recorded in cones and rods, and measurement of the glutamate transporter currents in rod- and cone recordings. These measurements suggest that Syt1 mediates all glutamate release from cones, and a fast component of glutamate release from rods.

Essential revisions:

Generally the reviewers were quite positive about the work and agree that it addresses the identity of the calcium sensor in rods and cones, a longstanding question in the field. Generally the data are of high quality and the manuscript is well presented. However, the following points should be addressed during revision.

1) Calcium cooperativity:

The data on Ca^2+^cooperativity is somehow detached from the main set of experiments. Figure 1 shows an experiment in which the apparent Ca^2+^cooperativity was determined in a fiber stimulation experiment in horizontal cells by means of varying the external Ca^2+^ concentration (the assumption was that release from cones was measured). A cooperativity of 2.7 was found and classified as "low". However, the "high" cooperativity of "4-5" cited by the authors for "many CNS synapses" has not been determined with similar methods (these were measured at the calyx of Held with Ca^2+^uncaging methods). Thus, it is unclear whether the present value of 2.7 can be directly compared to "4-5", and the reviewers feel that your conclusion about the Ca^2+^cooperativity at the cones being "low" is not sufficiently supported. Furthermore, the authors imply that Syt1 operates with a cooperativity of 2.7 in cones, but no data for the cooperativity in the absence of Syt1 is shown. For substantiating this conclusion, many more experiments are needed. Since such a thorough analysis would probably be beyond the scope of this work, we suggest to remove the claims on cooperativity, and the data of Figure 1 altogether – they are not essential for the main conclusions of the manuscript.

2) Morphological data:

The staining for syt1 (Figure 2C) in the OPL is difficult to understand for a non-specialist. PNA appears not to be explained. Also, it would be nice to see a rod-specific counterstain, because in Figure 2C middle the rod terminals are not visible, and so it is not possible to verify whether they express syt-1 or not. In Figure 2C right the conclusion is based on the absence of syt-1 staining within some drawn boundaries, but it is unclear how these boundaries were drawn. Additional explanatory labeling (using e.g. arrows etc.) may be helpful. A similar problem applies to the data shown in Figures 9-12 which are not intuitive and which may benefit from some condensation. For example, the CtBP2 intensities are much lower in both the rod- and the cone Syt1 cKO (Figure 10, second column) – does this have a meaning, or is it just a variability of antibody efficiency?

3) Measurement of release:

The methods applied here are quite indirect and it would be beneficial if the authors could add evidence from more direct measures. Is it possible to measure presynaptic capacitance changes, or release in paired recordings between rod / cones and a postsynaptic neuron? If yes, such measurements would significantly strengthen the paper.

In case such measurements are not feasible, the conclusions about fast and slow release, and about Ca^2+^-cooperativity (see also points 2, 4) should be tempered.

4) Discussion:

The Discussion should be more comprehensive and also mention other ribbon synapses that operate via graded membrane potential changes, such as hair cells, which also have a low-cooperativity in the Ca^2+^-dependent release that is nearly linear in mature mammalian and amphibian species (see Johnson et al., JNeurosci., 2008 (their Figure 1-3) and Cho et al., JNeurosci., 2011 (their Figure 6C)). It may be interesting to point out that otoferlin (the putative Ca-sensor for release) has multiple Ca^2+^-binding C2 domains and yet the release dependence on external Ca^2+^ is linear. Also, as mentioned above, the conclusions in many parts of the paper should be toned down, and re-calibrated given that quite indirect measures of release were employed.

---

## [Author Response]

Essential revisions:Generally the reviewers were quite positive about the work and agree that it addresses the identity of the calcium sensor in rods and cones, a longstanding question in the field. Generally the data are of high quality and the manuscript is well presented. However, the following points should be addressed during revision.1) Calcium cooperativity:The data on Ca^2+^cooperativity is somehow detached from the main set of experiments. Figure 1 shows an experiment in which the apparent Ca^2+^cooperativity was determined in a fiber stimulation experiment in horizontal cells by means of varying the external Ca^2+^ concentration (the assumption was that release from cones was measured). A cooperativity of 2.7 was found and classified as "low". However, the "high" cooperativity of "4-5" cited by the authors for "many CNS synapses" has not been determined with similar methods (these were measured at the calyx of Held with Ca^2+^uncaging methods). Thus, it is unclear whether the present value of 2.7 can be directly compared to "4-5", and the reviewers feel that your conclusion about the Ca^2+^cooperativity at the cones being "low" is not sufficiently supported. Furthermore, the authors imply that Syt1 operates with a cooperativity of 2.7 in cones, but no data for the cooperativity in the absence of Syt1 is shown. For substantiating this conclusion, many more experiments are needed. Since such a thorough analysis would probably be beyond the scope of this work, we suggest to remove the claims on cooperativity, and the data of Figure 1 altogether – they are not essential for the main conclusions of the manuscript.

As requested, we removed the data on cooperativity. As a consequence, we also rewrote the Introduction and Discussion to focus more on the impact of Syt1 removal on release and light-evoked ERG responses. In particular, results showing that Syt1 mediates the vast majority of release suggests that the kinetics of vesicle delivery down the ribbon, rather than use of different Ca^2+^ sensors, are more important in shaping release at photoreceptor ribbon synapses.

2) Morphological data:The staining for syt1 (Figure 2C) in the OPL is difficult to understand for a non-specialist. PNA appears not to be explained. Also, it would be nice to see a rod-specific counterstain, because in Figure 2C middle the rod terminals are not visible, and so it is not possible to verify whether they express syt-1 or not. In Figure 2C right the conclusion is based on the absence of syt-1 staining within some drawn boundaries, but it is unclear how these boundaries were drawn. Additional explanatory labeling (using e.g. arrows etc.) may be helpful. A similar problem applies to the data shown in Figures 9-12 which are not intuitive and which may benefit from some condensation. For example, the CtBP2 intensities are much lower in both the rod- and the cone Syt1 cKO (Figure 10, second column) – does this have a meaning, or is it just a variability of antibody efficiency?

We redid staining for Syt1 in Figure 1 (old Figure 2) to include an antibody to PSD95. In the retina, this antibody labels presynaptic photoreceptor terminals rather than post-synaptic proteins. As before, we co-labeled these sections with fluorescently-conjugated PNA to label the base of cone terminals and an antibody to Syt1. To help readers more readily appreciate Figures 9-12, we added a diagram to Figure 1 illustrating the different labels that we used to identify different cell types and structures (Figure 1C). At recent conferences, there has been considerable interest in these results as they pertain to synapse development and maintenance and so we thought it useful to show the changes (or lack thereof) in all three cell types (ON bipolar, OFF bipolar, and horizontal cells).

The ribbon protein Ribeye is an alternate transcript of the transcription factor, CtBP2, with both a ribbon-specific A-domain and B-domain shared by nuclear transcripts of CtBP2. Antibodies to CtBP2 used to label Ribeye at synaptic ribbons therefore also label CtBP2 in the nucleus. We adjusted CtBP2 signal intensity to maximize visibility of the ribbons. As we now state in the revised text, there were some differences among preparations in nuclear staining for CtBP2 but these did not appear to be systematically related to the mouse phenotype.

3) Measurement of release:The methods applied here are quite indirect and it would be beneficial if the authors could add evidence from more direct measures. Is it possible to measure presynaptic capacitance changes, or release in paired recordings between rod / cones and a postsynaptic neuron? If yes, such measurements would significantly strengthen the paper.In case such measurements are not feasible, the conclusions about fast and slow release, and about Ca^2+^-cooperativity (see also points 2, 4) should be tempered.

We have so far not had success with paired recordings from mouse photoreceptors and second-order neurons. We have had some success in capacitance recordings from cones but there is considerable cell to cell variability. In addition to the challenges of recording from these small cells, capacitance artefacts accompanying large depolarization-evoked conductance changes often confound cone capacitance changes. These artefacts are particularly prominent with long depolarizing steps, so capacitance measurements do not yet provide a reliable means for measuring slow release. With more time and experience, we hope to figure out methods for making cone capacitance recording a reliable measure for release but have not so far succeeded. We have modified the text to qualify the conclusions about slow release and have eliminated data on calcium cooperativity as suggested.

4) Discussion:The Discussion should be more comprehensive and also mention other ribbon synapses that operate via graded membrane potential changes, such as hair cells, which also have a low-cooperativity in the Ca-dependent release that is nearly linear in mature mammalian and amphibian species (see Johnson et al., JNeurosci., 2008 (their Figure 1-3) and Cho et al., JNeurosci., 2011 (their Figure 6C)). It may be interesting to point out that otoferlin (the putative Ca-sensor for release) has multiple Ca^2+^-binding C2 domains and yet the release dependence on external Ca^2+^ is linear. Also, as mentioned above, the conclusions in many parts of the paper should be toned down, and re-calibrated given that quite indirect measures of release were employed.

As recommended, we revised the Introduction and Discussion to remove most of the discussion surrounding cooperativity and Ca^2+^ sensitivity of photoreceptor release. We retained a brief discussion of the Ca^2+^ dependence of release from photoreceptors and include a discussion of the role of Syt4 in linearizing release from hair cells (Discussion, first paragraph). We qualify our results on slow release from rods by suggesting that it is possible there might be some residual Syt1 in rods from mice that could support a small amount of release (Discussion, subsection “Other possible Ca^2+^ sensors in photoreceptors”). Further studies will be needed to confirm this. One key conclusion is that almost all of the evoked release involves Syt1 suggesting that this sensor supports both phasic release evoked by brief steps but also sustained release in darkness that supports the ERG b-wave. This contrasts with synapses where slow release involves a different sensor altogether.